# ENERGY TRANSFORMER

## ABSTRACT

Transformers have become the de facto models of choice in machine learning, typically leading to impressive performance on many applications. At the same time, the architectural development in the transformer world is mostly driven by empirical findings, and the theoretical understanding of their architectural building blocks is rather limited. In contrast, Dense Associative Memory models or Modern Hopfield Networks have a well-established theoretical foundation, but have not yet demonstrated truly impressive practical results. We propose a transformer architecture that replaces the sequence of feedforward transformer blocks with a single large Associative Memory model. Our novel architecture, called Energy Transformer (or ET for short), has many of the familiar architectural primitives that are often used in the current generation of transformers. However, it is not identical to the existing architectures. The sequence of transformer layers in ET is purposely designed to minimize a specifically engineered energy function, which is responsible for representing the relationships between the tokens. As a consequence of this computational principle, the attention in ET is different from the conventional attention mechanism. In this work, we introduce the theoretical foundations of ET, explore it's empirical capabilities using the image completion task, and obtain strong quantitative results on the graph anomaly detection task.

## 1 INTRODUCTION

Transformers have become pervasive models in various domains of machine learning, including language, vision, and audio processing. Every transformer block uses four fundamental operations: attention, feed-forward multi-layer perceptron (MLP), residual connection, and layer normalization. Different variations of transformers result from combining these four operations in various ways. For instance, Press et al. (2019) propose to frontload additional attention operations and backload additional MLP layers in a sandwich-like instead of interleaved way, Lu et al. (2019) prepend an MLP layer before the attention in each transformer block, So et al. (2019) use neural architecture search methods to evolve even more sophisticated transformer blocks, and so on. Various methods exist to approximate the attention operation, multiple modifications of the norm operation, and connectivity of the block; see, for example, (Lin et al., 2021) for a taxonomy of different models. At present, however, the search for new transformer architectures is driven mostly by empirical evaluations, and the theoretical principles behind this growing list of architectural variations is missing.

Additionally, the computational role of the four elements remains the subject of discussions. Originally, Vaswani et al. (2017) emphasized attention as the most important part of the transformer block, arguing that the learnable long-range dependencies are more powerful than the local inductive biases of convolutional networks. On the other hand more recent investigations (Yu et al., 2021) argue that the entire transformer block is important. The "correct" way to combine the four basic operations inside the block remains unclear, as does an understanding of the core computational function of the entire block and each of its four elements.

In a seemingly unrelated line of work, Associative Memory models, also known as Hopfield Networks (Hopfield, 1982; 1984), have been gaining popularity in the machine learning community thanks to theoretical advancements pertaining to their memory storage capacity and novel architectural modifications. Specifically, it has been shown that increasing the sharpness of the activation functions can lead to super-linear (Krotov & Hopfield, 2016) and even exponential (Demircigil et al., 2017) memory storage capacity for these models, which is important for machine learning applications. This new class of Hopfield Networks is called Dense Associative Memories or Modern

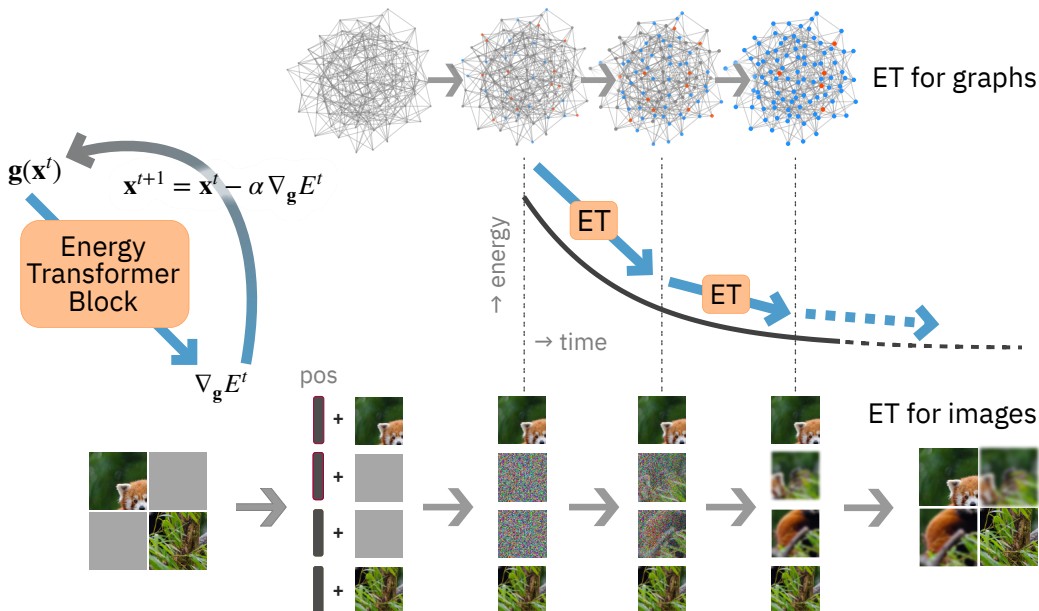

Figure 1: Overview of the Energy Transformer (ET). Instead of a sequence of conventional transformer blocks, a single recurrent ET block is used. The operation of this block is dictated by the global energy function. The token representations are updated according to a continuous time differential equation with the time-discretized update step $\alpha = dt/\tau$. On the image domain, images are split into non-overlapping patches that are linearly encoded into tokens with added learnable positional embeddings (POS). Some patches are randomly masked. These tokens are recurrently passed through ET, and each iteration reduces the energy of the set of tokens. The token representations at or near the fixed point are then decoded using the decoder network to obtain the reconstructed image. The network is trained by minimizing the mean squared error loss between the reconstructed image and the original image. On the graph domain, the same general pipeline is used. Each token represents a node, and each node has its own positional encoding. The token representations at or near the fixed point are used for the prediction of the anomaly status of each node.

**Hopfield Networks.** Ramsauer et al. (2020) additionally describe how the attention mechanism in transformers is closely related to a special model of this family with the softmax activation function.

There are high-level conceptual similarities between transformers and Dense Associative Memories, since both architectures are designed for some form of denoising of the input. Transformers are typically pre-trained on a masked-token task, e.g., in the domain of Natural Language Processing (NLP) certain tokens in the sentence are masked and the model predicts the masked tokens. Dense Associative Memory models are designed for completing the incomplete patterns. For instance, a pattern can be the concatenation of an image and its label, and the model can be trained to predict part of the input (the label), which is masked, given the query (the image). They can also be trained in a self-supervised way by predicting the occluded parts of the image, or denoising the image.

There are also high-level differences between the two classes of models. Associative Memories are recurrent networks with a global energy function so that the network dynamics converges to a fixed point attractor state corresponding to a local minimum of the energy function. Transformers are typically not described as dynamical systems at all. Rather, they are thought of as feed-forward networks built of the four computational elements discussed above. Even if one thinks about them as dynamical systems with tied weights, e.g., (Bai et al., 2019), there is no reason to expect that their dynamics converge to a fixed point attractor (see the discussion in (Lan et al., 2020)).

Additionally, a recent study (Yang et al., 2022) uses a form of Majorization-Minimization algorithms (Sun et al., 2016) to interpret the forward path in the transformer block as an optimization process. This interpretation requires imposing certain constraints on the operations inside the block, and attempting to find an energy function that describes the constrained block. We take a complementary

approach by using intuition developed in Associative Memory models to *start* with an energy function that is perfectly suited for the problem of interest. The optimization process and the resulting transformer block in our approach is a *consequence* of this specifically chosen energy function.

Concretely, we use the recent theoretical advancements and architectural developments in Dense Associative Memories to design an energy function tailored to route the information between the tokens. The goal of this energy function is to represent the relationships between the semantic contents of tokens describing a given data point (e.g., the relationships between the contents of the image patches in the vision domain, or relationships between the nodes' attributes in the graph domain). The core mathematical idea of our approach is that the sequence of these unusual transformer blocks, which we call the Energy Transformer (ET), minimizes this global energy function. Thus, the sequence of conventional transformer blocks is replaced with a single ET block, which iterates the token representations until they converge to a fixed point attractor state. In the image domain, this fixed point corresponds to the completed image with masked tokens replaced by plausible auto-completions of the occluded image patches. In the graph domain, the fixed point reveals the anomaly status of a given node given that node's neighbors; see Figure 1. The energy function in our ET block is designed with the goal to describe the *relationships between the tokens*. Examples of relationships in the image domain are: straight lines tend to continue through multiple patches, given a face with one eye being masked the network should impaint the missing eye, etc. In the graph domain, these are the relationships between the attributes and the anomaly status of the connected nodes. The optimization procedure during the forward path of ET uses continuous time differential equations, and describes a gradient decent on the specifically chosen energy function.

The core mathematical principle of the ET block – the existence of the global energy function – dictates strong constraints on the possible operations, the order in which these operations are executed in the forward path, and the symmetries of the weights in the network. As a corollary of this theoretical principle, the attention mechanism of ET is different from the attention mechanism commonly used in feed-forward transformers (Vaswani et al., 2017).

In the following section we introduce the global energy function for the ET block and explain the block's architecture. We then explore the inner workings of the ET network for image completion and qualitatively assess the learned representations (the model is trained on ImageNet-1k in a general pipeline similar to Dosovitskiy et al. (2021)). Finally, we turn to the graph anomaly detection task, which is conceptually similar to the image completion setting but has a record of strong published benchmarks against which our approach can be quantitatively compared. We show that the ET network stands in line with or outperforms the latest benchmarks. Although we focus on the computer vision and anomaly detection domains in this paper, we believe that the computational principles developed can be applied to other exciting domains (e.g., NLP, audio, and video) in which conventional transformers have shown promising results.

## 2 ENERGY TRANSFORMER BLOCK

We now introduce the theoretical framework of the ET network. For clarity of presentation, we use language associated with the image domain. For the graph domain, one should think about "image patches" as nodes on the graph.

The overall pipeline is similar to the Vision Transformer networks (ViTs) and is shown in Figure 1. An input image is split into non-overlapping patches. After passing these patches through the encoder and adding the positional information, the semantic content of each patch and its position is encoded in the token $x_{iA}$. In the following the indices $i, j, k = 1...D$ are used to denote the token vector's elements, indices $A, B, C = 1...N$ are used to enumerate the patches and their corresponding tokens. It is helpful to think about each image patch as a physical particle, which has a complicated internal state described by a $D$-dimensional vector $\mathbf{x_A}$. This internal state describes the identity of the particle (representing the pixels of each patch), and the particle's positional embedding (the patch's location within the image). The ET block is described by a continuous time differential equation, which describes interactions between these particles. Initially, at $t = 1$ the network is given a set containing two groups of particles corresponding to open and masked patches. The "open" particles know their identity and location in the image. The "masked" particles only know where in the image they are located, but are not provided the information about what image patch they represent. The goal of ET's non-linear dynamics is to allow the masked particles to find

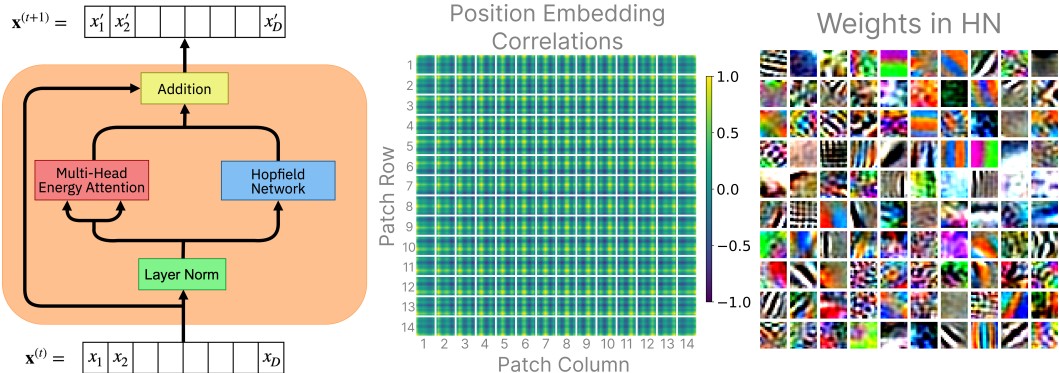

Figure 2: **Left**: Inside the ET block. The input token **x** passes through a sequence of operations and gets updated to produce the output token **x**′. The operations inside the ET block are carefully engineered so that the entire network has a global energy function, which decreases with time and is bounded from below. In contrast to conventional transformers, the ET-based analogs of the attention module and the feed-forward MLP module are applied in parallel as opposed to consecutively. **Center**: The cosine similarity between the learned position embedding of each patch and every other patch. In each cell, the brightest patch indicates the cell of consideration. **Right**: 100 selected memories stored in the HN memory matrix, visualized by the decoder as 16x16 RGB image patches. This visualization is unique to our model, as traditional Transformers cannot guarantee image representations in the learned weights.

an identity consistent with their locations and the identities of open particles. This dynamical evolution is designed so that it minimizes a global energy function, and is guaranteed to arrive at a fixed point attractor state. The identities of the masked particles are considered to be revealed when the dynamical trajectory reaches the fixed point. Thus, the central question is: how can we design the energy function that accurately captures the task that the Energy Transformer needs to solve?

The masked particles' search for identity is guided by two pieces of information: identities of the open particles, and the general knowledge about what patches are in principle possible in the space of all possible images. These two pieces of information are described by two contributions to the ET's energy function: the energy based attention and the Hopfield Network, respectively, for reasons that will become clear in the next sections. Below we define each element of the ET block in the order they appear in Figure 2.

LAYER NORM

Each token is represented by a vector $\mathbf{x} \in R^D$. At the same time, most of the operations inside the ET block are defined using a layer-normalized token representation

$$g_i = \gamma \frac{x_i - \bar{x}}{\sqrt{\frac{1}{D} \sum_j \left(x_j - \bar{x}\right)^2 + \varepsilon}} + \delta_i, \quad \text{where} \quad \bar{x} = \frac{1}{D} \sum_{k=1}^{D} x_k \tag{1}$$

The scalar $\gamma$ and the vector $\delta_i$ are learnable parameters, $\varepsilon$ is a small regularization constant. Importantly, this operation can be viewed as an activation function for the neurons and can be defined as a partial derivative of the Lagrangian function

$$L = D\gamma \sqrt{\frac{1}{D} \sum_j \left(x_j - \bar{x}\right)^2 + \varepsilon} + \sum_j \delta_j x_j, \quad \text{so that} \quad g_i = \frac{\partial L}{\partial x_i} \tag{2}$$

See Krotov & Hopfield (2021); Tang & Kopp (2021); Krotov (2021) for a detailed discussion of this property.

MULTI-HEAD ENERGY ATTENTION

The first contribution to the ET's energy function is responsible for exchanging information between the particles (patches). Similarly to the conventional attention mechanism, each token generates a pair of queries and keys (ET does not have a separate value matrix; instead the value matrix is a function of keys and queries). The goal of the energy based attention is to evolve the tokens in such a way that the keys of the open patches are aligned with the queries of the masked patches in the internal space of the attention operation. Below we use index $\alpha = 1...Y$ to denote elements of this internal space, and index $h = 1...H$ to denote different heads of this operation. With these notations the energy-based attention operation is described by the following energy function:

$$E^{\text{ATT}} = -\frac{1}{\beta} \sum_h \sum_C \log \left( \sum_{B \neq C} \exp \left( \beta \sum_\alpha K_{\alpha h B} \, Q_{\alpha h C} \right) \right) \tag{3}$$

where the queries and keys tensors are defined as

$$K_{\alpha h B} = \sum_j W^K_{\alpha h j} \, g_{jB}, \qquad \mathbf{K} \in R^{Y \times H \times N}$$
$$Q_{\alpha h C} = \sum_j W^Q_{\alpha h j} \, g_{jC}, \qquad \mathbf{Q} \in R^{Y \times H \times N} \tag{4}$$

and the tensors $\mathbf{W}^K \in R^{Y \times H \times D}$ and $\mathbf{W}^Q \in R^{Y \times H \times D}$ are learnable parameters.

From the computational perspective each patch generates two representations: query (given the position of the patch and its current content, where in the image should it look for the prompts on how to evolve in time?), and key (given the current content of the patch and its position, what should be the contents of the patches that attend to it?). The log-sum energy function (3) is minimal when for every patch in the image its queries are aligned with the keys of a small number of other patches connected by the attention map. Different heads (index $h$) contribute to the energy additively.

HOPFIELD NETWORK MODULE

The next step of the ET block, which we call the Hopfield Network (HN), is responsible for ensuring that the token representations are consistent with what one expects to see in realistic images. The energy of this sub-block is defined as:

$$E^{\text{HN}} = -\frac{1}{2} \sum_{B,\mu} r \left( \sum_j \xi_{\mu j} \, g_{jB} \right)^2, \qquad \xi \in R^{K \times D} \tag{5}$$

where $\xi_{\mu j}$ is a set of learnable weights (memories in the Hopfield Network), and $r(\cdot)$ is an activation function. Depending on the choice of the activation function this step can be viewed either as a classical continuous Hopfield Network (Hopfield, 1984) if the activation function grows slowly (e.g., ReLU), or as a modern continuous Hopfield Network (Krotov & Hopfield, 2016; Ramsauer et al., 2020; Krotov & Hopfield, 2021) if the activation function is sharply peaked around the memories (e.g. power or softmax). The HN sub-block is analogous to the feed-forward MLP step in the conventional transformer block but requires that the weights of the projection from the token space to the hidden neuron's space to be the same (transposed matrix) as the weights of the subsequent projection from the hidden space to the token space. Thus, the HN module here is an MLP with shared weights that is *applied recurrently*. The energy contribution of this block is low when the tokens representations are aligned with some rows of the matrix $\xi$, which represent memories.

DYNAMICS OF TOKEN UPDATES

The forward path of the ET network is described by the continuous time differential equation, which minimizes the sum of the two energies described above

$$\tau \frac{dx_{iA}}{dt} = -\frac{\partial E}{\partial g_{iA}}, \quad \text{where} \quad E = E^{\text{ATT}} + E^{\text{HN}} \tag{6}$$

Here $x_{iA}$ is the token representation (input and output from the ET block), and $g_{iA}$ is its layer-normalized version. The first energy is low when each patch's queries are aligned with the keys

of its neighbors. The second energy is low when each patch has content consistent with the general expectations about what an image patch should look like (memory slots of the matrix $\chi$). The dynamical system (6) finds a trade-off between these two desirable properties of each token's representation. For numerical evaluations equation (6) is discretized in time.

To demonstrate that the dynamical system (6) minimizes the energy, consider the temporal derivative

$$\frac{dE}{dt} = \sum_{i,j,A} \frac{\partial E}{\partial g_{iA}} \frac{\partial g_{iA}}{\partial x_{jA}} \frac{dx_{jA}}{dt} = -\frac{1}{\tau} \sum_{i,j,A} \frac{\partial E}{\partial g_{iA}} M_{ij}^A \frac{\partial E}{\partial g_{jA}} \leq 0 \tag{7}$$

The last inequality sign holds if the symmetric part of the matrix

$$M_{ij}^A = \frac{\partial g_{iA}}{\partial x_{jA}} = \frac{\partial^2 L}{\partial x_{iA} \partial x_{jA}} \tag{8}$$

is positive semi-definite (for each value of index $A$). The Lagrangian (2) satisfies this condition.

RELATIONSHIP TO MODERN HOPFIELD NETWORKS AND CONVENTIONAL ATTENTION

One of the theoretical contributions of our work is the design of the energy attention mechanism and the corresponding energy function (3). Although heavily inspired by prior work on Modern Hopfield Networks, our approach is fundamentally different from it. Our energy function (3) may look somewhat similar to the energy function of a continuous Hopfield Network with the softmax activation function. The main difference, however, is that in order to use Modern Hopfield Networks recurrently (as opposed to applying their update rule only once) the keys must be constant parameters (called memories in the Hopfield language). In contrast, in our energy attention network the keys are dynamical variables that evolve in time with the queries.

To emphasize this further, it is instructive to write explicitly the ET attention contribution to the update dynamics (6). It is given by (for clarity, assume only one head of attention):

$$-\frac{\partial E^{\text{ATT}}}{\partial g_{iA}} = \sum_{C \neq A} \sum_{\alpha} W_{\alpha i}^Q K_{\alpha C} \underset{C}{\text{softmax}} \left( \beta \sum_{\gamma} K_{\gamma C} Q_{\gamma A} \right) + W_{\alpha i}^K Q_{\alpha C} \underset{A}{\text{softmax}} \left( \beta \sum_{\gamma} K_{\gamma A} Q_{\gamma C} \right)$$

In both terms the softmax normalization is done over the token index of the keys, which is indicated by the subscript in the equation. The first term in this formula is the conventional attention mechanism (Vaswani et al., 2017) with the value matrix equal to $\mathbf{V} = (\mathbf{W}^Q)^T \mathbf{K} = \sum_{\alpha} W_{\alpha i}^Q K_{\alpha C}$. The second term is the brand new contribution that is missing in the original attention mechanism. The presence of this second term is crucial to make sure that the dynamical system (6) minimizes the energy function if applied recurrently. This second term is the main difference of our approach compared to the Modern Hopfield Networks. The same difference applies to the other recent proposals (Yang et al., 2022).

Lastly, we want to emphasize that our ET block contains two different kinds of Hopfield Networks acting in parallel, see Figure 2. The first one is the energy attention module, which is inspired by, but not identical to, Modern Hopfield Networks. The second one is the "Hopfield Network" module, which can be either a classical or modern Hopfield Network. These two should not be confused.

## 3 QUALITATIVE INSPECTION OF THE ET FRAMEWORK ON IMAGENET

We have trained the ET network on the masked image completion task using ImageNet-1k dataset (Deng et al., 2009). Each image was broken into non-overlapping patches of 16x16 RGB pixels, which were projected with a single affine encoder into the token space. Half of these tokens were "masked", e.g., by replacing them with a learnable MASK token. A distinct learnable position encoding vector was added to each token. Our ET block then processes all tokens recurrently for $T$ steps. The token representations after $T$ steps are passed to a simple linear decoder (consisting of a layer norm and an affine transformation). The loss function is the standard MSE loss on the occluded patches. See more details on the implementation and the hyperparameters in Appendix A.

Examples of occluded/reconstructed images (unseen during training) are shown in Figure 3. In general, our model learns to perform the task very well, capturing the texture in dog fur (col 3) and

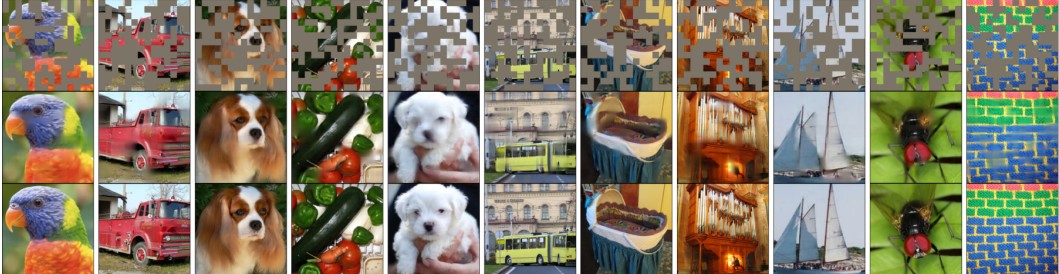

Figure 3: Reconstruction examples of our Energy Transformer using images from the ImageNet-1k validation set. *Top row:* input images where 50% of the patches are masked with the learned MASK token. *Middle row*: output reconstructions after 12 time steps. *Bottom row*: original images.

understanding meaningful boundaries of objects. However, we observe that our single ET block struggles to understand some global structure, e.g., failing to capture both eyes of the white dog (col 4) and completing irregular brick patterns in the name of extending the un-occluded borders (last col). We additionally inspect the positional encoding vectors associated with every token, Figure 2, where the model learns a locality structure in the image plane that is very similar to the original ViT (Dosovitskiy et al., 2021). The position embedding of each image patch has learned high similarity values to other patches in the same row and column, with similarity values higher for neighboring tokens than distant tokens.

Our network is unique compared to standard ViTs in that the iterative dynamics only *move* tokens around in the same space from which the final fixed point representation can be decoded back into the image plane. This functionality makes it possible to visualize essentially any *token representation*, *weight*, or *gradient of the energy* directly in the image plane. This feature is highly desirable from the perspective of interpretability, since it makes it possible to track the updates performed by the network directly in the image plane as the computation unfolds in time. In Figure 2 this functionality is used for inspecting the learned weights of the HN module directly in the image plane. According to our theory, these weights should represent basis vectors in the space of all possible image patches. These learned representations look qualitatively similar to the representations typically found in networks trained on image datasets, e.g., Zeiler & Fergus (2014).

We additionally visualize the gradients of the energy function (which are equal to the token updates, see Equation 6) of both ATTN block and the HN block, see Figure 4. Early in time, almost all signal to the masked tokens comes from the ATTN block, which routes information from the open patches to the masked ones; no meaningful signal comes from the HN block to the masked patch dynamics. Later in time we observe a different phenomenon: almost all signal to masked tokens comes from the HN module while ATTN contributes a blurry and uninformative signal. Thus, the attention layer is crucial early in the network dynamics, feeding signal to masked patches from the visible patches, whereas the HN is crucial later in the dynamics as the model approaches the final reconstruction, sharpening the masked patches. All the qualitative findings presented in this section are in accord with the core computational strategy of the ET block as it was designed theoretically in section 2.

## 4 GRAPH ANOMALY DETECTION

Having built the theoretical foundation of the ET network and gained an intuition about its inner workings through visualizations, we turn to quantitatively evaluating its performance on the graph anomaly detection problem, a task with plenty of strong and recently published baselines. Anomalies are outliers that significantly deviate in their properties from the majority of the samples. Detecting anomalies on graphs has broad applications in cybersecurity (Hong et al., 2014; Pan et al., 2019), fraud detection (Huang et al., 2018; Pourhabibi et al., 2020), and social networks (Chaker et al., 2017). Generally, there are three types of graph anomalies: node anomaly, edge anomaly, and subgraph anomaly. In this work, we focus on node anomaly detection in attributed graphs. This task is perfectly suited for the ET network, since each node's attributes can be encoded in the latent

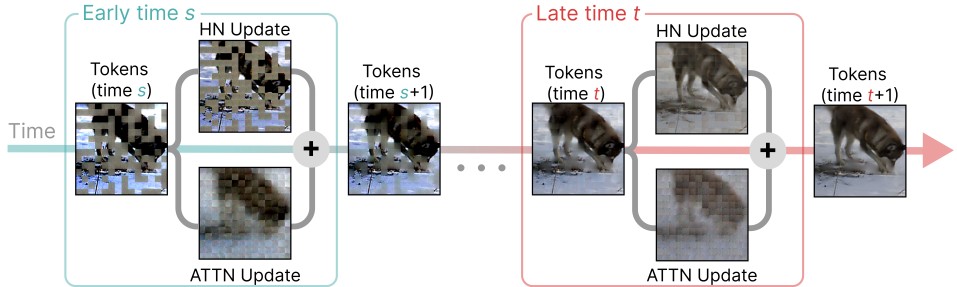

Figure 4: Token representations and gradients are visualized using the decoder at different times during the dynamics. The Energy Attention (ATTN) block contributes general structure information to the masked patches at *earlier* time steps, whereas the Hopfield Network (HN) significantly sharpens the quality of the masked patches at *later* time steps.

space and treated as a token (with added learnable positional embeddings). The network iterates these representations in time, and the outputs can be used for the node anomaly classification task.

Graph Convolutional Networks (GCN) (Kipf & Welling, 2016) have been widely used for this task due to their capability of learning high level representations of graph structures and node attributes (Ding et al., 2019; Peng et al., 2020). However, vanilla GCNs suffer from the over-smoothing problem (Wu et al., 2019). In each layer of the forward pass, the outlier node aggregates information from its neighbors. This averaging makes the features of anomalies less distinguishable from the features of benign nodes. Our approach does not suffer from this problem, since the routing of the information between the nodes is done through the energy based attention, which uses different aggregation procedure depending on whether or not the node is anomalous.

In order to turn the anomaly detection task on graphs into the ET framework, consider an undirected graph with $N$ nodes. Every node has a vector of attributes $\mathbf{y}_A \in R^F$, where $F$ is the number of node's features. Additionally, every node has a binary label $l_A$, indicating whether the node is benign or not. We focus on node anomaly and assume that all edges are trusted. The task is to predict the label of the node given the graph structure and the node's features. Since there are far more benign nodes in the graph than anomalous ones, anomaly detection can be regarded as an imbalanced node classification task.

First, the feature vectors for every node are converted to a token representation using a linear embedding $\mathbf{E}$ and adding a learnable positional embedding $\lambda_A$

$$\mathbf{x}_A^{t=1} = \mathbf{E}\mathbf{y}_A + \lambda_A \tag{9}$$

where the superscript $t = 1$ indicates the time of the update of the ET dynamics. This token representation is iterated through the ET block for $T$ iterations. When the retrieval dynamics becomes stable, we have the final representation for each node $\mathbf{x}_A^{t=T}$ (or more precisely $\mathbf{g}_A^{t=T}$, since the outputs are additionally passed through a layer norm operation after the final ET update). This output is concatenated with the initial (layer normalized) token to form the final output of the network

$$\mathbf{g}_A^{\text{final}} = \mathbf{g}_A^{t=1} \,||\, \mathbf{g}_A^{t=T} \tag{10}$$

Following Tang et al. (2022), the node representation $\mathbf{g}_A^{\text{final}}$ is fed into an MLP with the sigmoid activation function to compute the anomaly probabilities $p_A$. The weighted cross entropy

$$\text{Loss} = \sum_A \left[ \sigma \, l_A \log(p_A) + (1 - l_A) \log(1 - p_A) \right] \tag{11}$$

is used to train the whole network. Above, $\sigma$ is the ratio of the regular labels ($l_A = 0$) to anomalous labels ($l_A = 1$).

### 4.1 EXPERIMENTAL EVALUATION

Four datasets are used for the graph anomaly detection experiments. YelpChi dataset (Rayana & Akoglu, 2015) aims at opinion spam detection in Yelp reviews. Amazon dataset is used to detect

anomalous users under the Musical Instrument Category on *amazon.com* (McAuley & Leskovec, 2013). T-Finance and T-Social datasets (Tang et al., 2022) are used for anomalous account detection in the transactions and social networks, respectively. For these four datasets, the graph is treated as a homogeneous graph (i.e. all the edges are of the same type), and a feature vector is associated with each node. The task is to predict the label (anomaly status) of the nodes. For each dataset, either 1% or 40% of the nodes are used for training, and the remaining 99% or 60% are split 1 : 2 into validation and testing, see Appendix B for details.

We compare with state-of-the-art approaches for graph anomaly detection, which include Graph-Consis (Liu et al., 2020), CAREGNN (Dou et al., 2020), PC-GNN (Liu et al., 2021) and BWGNN (Tang et al., 2022). Additionally, multi-layer perceptrons (MLP) and Graph Transformer (GT) (Dwivedi & Bresson, 2020) are included in the baselines for completeness. Following previous work, macro-F1 score (unweighted mean of F1 score) and the Area Under the Curve (AUC) are used as the evaluation metrics on the test datasets Davis & Goadrich (2006). See Appendix B for more details on training protocols and the hyperparameters choices. The results are reported in Table 1. Our ET network demonstrates very strong results across all the datasets.

Table 1: Performance of all the methods on Yelp, Amazon, T-Finance, and T-Social datasets with different training ratios. Following Tang et al. (2022), mean and standard deviation over 5 runs with different train/dev/test split are reported for our method and the baselines (standard deviations are only included if they are available in the prior work). Best results are in **bold**. Our model is state of the art or near state of the art on every category.

| | Datasets | Split | GraphConsis | CAREGNN | PC-GNN | BWGNN | MLP | GT | ET (Ours) |
|---|---|---|---|---|---|---|---|---|---|
| **Macro-F1** | Yelp | 1% | $56.8_{\pm2.8}$ | $62.1_{\pm1.3}$ | $59.8_{\pm1.4}$ | $61.1_{\pm0.4}$ | $53.9_{\pm0.2}$ | $61.7_{\pm0.4}$ | $\mathbf{63.0_{\pm0.6}}$ |
| | | 40% | $58.7_{\pm2.0}$ | $63.3_{\pm0.9}$ | $63.0_{\pm2.3}$ | $71.0_{\pm0.9}$ | $57.5_{\pm0.8}$ | $68.7_{\pm0.4}$ | $\mathbf{71.5_{\pm0.1}}$ |
| | Amazon | 1% | $68.5_{\pm3.4}$ | $68.7_{\pm1.6}$ | $79.8_{\pm5.6}$ | $\mathbf{90.9_{\pm0.7}}$ | $74.6_{\pm1.2}$ | $88.6_{\pm0.5}$ | $89.3_{\pm0.7}$ |
| | | 40% | $75.1_{\pm3.2}$ | $86.3_{\pm1.7}$ | $89.5_{\pm0.7}$ | $92.2_{\pm0.4}$ | $79.1_{\pm1.2}$ | $91.7_{\pm0.8}$ | $\mathbf{92.8_{\pm0.3}}$ |
| | T-Finance | 1% | $71.7$ | $73.3$ | $62.0$ | $84.8$ | $61.0$ | $81.5$ | $\mathbf{85.1_{\pm1.0}}$ |
| | | 40% | $73.4$ | $77.5$ | $63.1$ | $86.8$ | $70.5$ | $83.6$ | $\mathbf{88.2_{\pm1.0}}$ |
| | T-Social | 1% | $52.4$ | $55.8$ | $51.1$ | $75.9$ | $50.0$ | $64.3$ | $\mathbf{79.1_{\pm0.7}}$ |
| | | 40% | $56.5$ | $56.2$ | $52.1$ | $\mathbf{83.9}$ | $50.3$ | $68.2$ | $83.5_{\pm0.4}$ |
| **AUC** | Yelp | 1% | $66.4_{\pm3.4}$ | $75.0_{\pm3.8}$ | $\mathbf{75.4_{\pm0.9}}$ | $72.0_{\pm0.5}$ | $59.8_{\pm0.4}$ | $72.5_{\pm0.6}$ | $73.2_{\pm0.8}$ |
| | | 40% | $69.8_{\pm3.0}$ | $76.1_{\pm2.9}$ | $79.8_{\pm0.1}$ | $84.0_{\pm0.9}$ | $66.5_{\pm1.0}$ | $81.9_{\pm0.5}$ | $\mathbf{84.9_{\pm0.3}}$ |
| | Amazon | 1% | $74.1_{\pm3.5}$ | $88.6_{\pm3.5}$ | $90.4_{\pm2.0}$ | $89.4_{\pm0.3}$ | $83.6_{\pm1.7}$ | $89.0_{\pm1.2}$ | $\mathbf{91.9_{\pm1.0}}$ |
| | | 40% | $87.4_{\pm3.3}$ | $90.5_{\pm1.6}$ | $95.8_{\pm0.1}$ | $98.0_{\pm0.4}$ | $89.8_{\pm1.0}$ | $95.4_{\pm0.6}$ | $\mathbf{97.3_{\pm0.4}}$ |
| | T-Finance | 1% | $90.2$ | $90.5$ | $90.7$ | $91.1$ | $82.9$ | $90.0$ | $\mathbf{92.8_{\pm1.1}}$ |
| | | 40% | $91.4$ | $92.1$ | $91.2$ | $94.3$ | $87.1$ | $88.2$ | $\mathbf{95.0_{\pm3.0}}$ |
| | T-Social | 1% | $65.2$ | $71.2$ | $59.8$ | $88.0$ | $56.3$ | $81.4$ | $\mathbf{91.9_{\pm0.6}}$ |
| | | 40% | $71.2$ | $71.8$ | $68.4$ | $\mathbf{95.2}$ | $56.9$ | $82.5$ | $93.9_{\pm0.2}$ |

## 5 DISCUSSION AND CONCLUSIONS

A lot of recent research has been dedicated to understanding the striking analogy between Hopfield Networks and the attention mechanism in transformers. At a high level, the main message of our work is that the *entire* transformer block (including feed-forward MLP, layer normalization, and residual connections) can be viewed as a single large Hopfield Network, not just attention alone. At a deeper level, we use recent advances in the field of Hopfield Networks to design a novel energy function that is tailored for dynamical information routing between the tokens and representation of a large number of relationships between those tokens. When used in the encoder-decoder setting, an appealing feature of our network is that any state, weight, or state update can be mapped directly into the data domain. This provides the possibility to inspect the inner workings of the whole network, contributing to its interpretability. The attention mechanism in our network contains an important extra term compared to conventional attention. We have tested the ET network on the image completion task (qualitatively) and node anomaly detection on graphs (quantitatively). The qualitative investigation reveals the perfect alignment between the theoretical design principles of our network and its empirical computation. The quantitative evaluation demonstrates strong results, which stand in line or exceed the methods recently developed specifically for this task. Although we have only tested ET on two tasks, we intentionally picked two entirely different data domains (images and graphs). We believe that the proposed network will be useful for other tasks and domains and deserves a comprehensive investigation in line with other popular variants of transformers.

## 6 REPRODUCIBILITY STATEMENT

*Section does not count towards page limit*

In the experiments presented in this paper we have taken several steps to ensure reproducibility of our results. Namely, all the training protocols and implementation details including the hyperparameter selection are described in Appendix A, Appendix B, and Appendix F. The model and training code is released anonymously.[1] The code for image reconstruction is written in JAX Bradbury et al. (2018) with a single entry script to launch the training process, with defaults set to the configuration that produced the models used in this paper. The training script sets a seed that can recreate the exact same training setup as we had, ensuring the exact same weight initialization and random data augmentation provided default arguments. No additional training data was used beyond the training set (ImageNet-1k 2012), which is publicly available. The code for Graph Anomaly Detection is written in PyTorch. Given the nature of the anomaly detection problem (random splits into training, validation, testing sets) all our results on graphs are reported with mean and standard deviations, describing typical variability in the performance.

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

## A    DETAILS OF TRAINING ON IMAGENET

We trained the ET network on a masked-image completion task on the ImageNet-1k (IN1K) dataset. We treat all images in IN1K as images of shape $224 \times 224$ that are normalized according to standard IN1K practices (mean 0, variance 1 on the channel dimension) and use data augmentations provided by the popular `timm` library (Wightman, 2019) (See Table 2). Following the conventional ViT pipeline (Dosovitskiy et al., 2021), we split these images into non-overlapping patches of 16x16 RGB pixels which are then projected with a single affine encoder into the token dimension $D$ for a total of 196 encoded tokens per image. We proceed to randomly and uniformly assign 100 of these tokens as "occluded" which are the only tokens considered by the loss function. "Occluded" tokens are designated as follows: of the 100 tokens, 90 tokens are replaced with a learnable MASK token of dimension $D$ and 10 we leave untouched (which we find important for the HN to learn meaningful patch representations). To all tokens we then add a distinct learnable position bias.

These tokens are then passed to our Energy Transformer block which we recur for $T$ steps (the "depth" of the model in conventional Transformers). At each step, the feedback signal (the sum of the energy gradients from our attention block and HN block) is subtracted from our original token representation with a scalar step size $\alpha = \frac{dt}{\tau}$ which we treat as a non-learnable hyperparameter in our experiments. The token representations after $T$ steps are passed to a simple linear decoder (consisting of a layer norm and an affine transformation) to project our representations back into the image plane. We then use the standard MSE Loss between the original pixels and reconstructed pixels for only the 100 occluded patches. We allow self attention as in the following formula for the energy of multiheaded attention.

$$E^{\text{ATT}} = \sum_h -\frac{1}{\beta} \sum_C \log \left( \sum_B \exp \left( \beta \sum_\alpha K_{\alpha h B} \, Q_{\alpha h C} \right) \right) \tag{12}$$

We give details of our architectural choices in Table 2. In the main paper we present our Energy Transformer with a configuration similar to the standard base Transformer configuration (e.g., token

dimension 768, 12 heads each with $Y = 64$, softmax's $\beta = \frac{1}{\sqrt{Y}}$, ...), with several considerations learned from the qualitative image evaluations:

- The $\frac{dt}{\tau}$ (step size) of 1 implicitly used in the traditional transformer noticeably degrades our ability to smoothly descend the energy function. We find that a step size of 0.1 provides a smoother descent down the energy function and benefits the image reconstruction quality.
- We observe that our MSE loss must include some subset of un-occluded patches in order for the HN to learn meaningful filters.
- Values of $\beta$ in the energy attention that are too high prevent our model from training. This is possibly due to vanishing gradients in our attention operation from a `softmax` operation that is too spiky.
- Without gradient clipping, our model fails to train at the learning rates we tried higher than 1e-4. We observe that gradient clipping not only helps our model train faster at the trainable learning rates, it also allows us to train at higher learning rates.

Our architecture and experiments for the image reconstruction task were written in JAX (Bradbury et al., 2018) using Flax (Heek et al., 2020). This engineering choice means that our architecture definitions are quite lightweight, as we can define the desired energy function of the ET and use JAX's autograd to automatically calculate the desired update. All training code and software will be released upon the paper's acceptance.

Table 2: Hyperparameter, architecture, and data augmentation choices for ET-base during ImageNet-1k masked training experiments. Data augmentations are listed as parameters passed to the equivalent `timm` dataloader functionality.

| Training | | Architecture | | Data Augmentation | |
|---|---|---|---|---|---|
| batch_size | 768 | token_dim | 768 | random_erase | None |
| epochs | 100 | num_heads | 12 | horizontal_flip | 0.5 |
| lr | 5e-4 | head_dim | 64 | vertical_flip | 0 |
| warmup_epochs | 2 | $\beta$ | 1/8 | color_jitter | 0.4 |
| start & end lr | 5e-7 | train_betas | No | scale | (0.08, 1) |
| b1, b2 (ADAM) | 0.9, 0.99 | step size $\alpha$ | 0.1 | ratio | (3/4, 4/3) |
| weight_decay | 0.05 | depth | 12 | auto_augment | None |
| grad_clipping | 1. | hidden_dim HN | 3072 | | |
| | | bias in HN | None | | |
| | | bias in ATT | None | | |
| | | bias in LNORM | Yes | | |

## A.1 EXPLORING THE HOPFIELD MEMORIES

A distinctive aspect of our network is that any variable that has a vector index $i$ of tokens can be mapped into the data domain by applying the decoder network to this variable. This makes it possible to inspect all the weights in the model. For instance, the concept of "memories" is crucial to understanding how Hopfield networks function. The memories within the HN module represent the building blocks of *all possible image patches* in our data domain, where an encoded image patch is a superposition of a subset of memories. The complete set of memory vectors from the HN module is shown in Figure 5. The same analysis can be applied to the weights of the ET-attention module. In Figure 6, we show all the weights from this module mapped into the image plane.

## A.2 BIAS CORRELATIONS

The relationships between our position embeddings exhibit similar behavior to the position correlations of the original ViT in that they are highly susceptible to choices of the hyperparameters (Figure 10 of Dosovitskiy et al. (2021)). In particular, we consider the effect of weight decay and the $\beta$ parameter that serves as the inverse temperature of the attention operation (see Equation 3). The lower the temperature (i.e., the higher the value of $\beta$), the spikier the softmax distribution. By using a lower $\beta$, we encourage the attention energy to distribute its positional embeddings across a wider range of patches in the model.

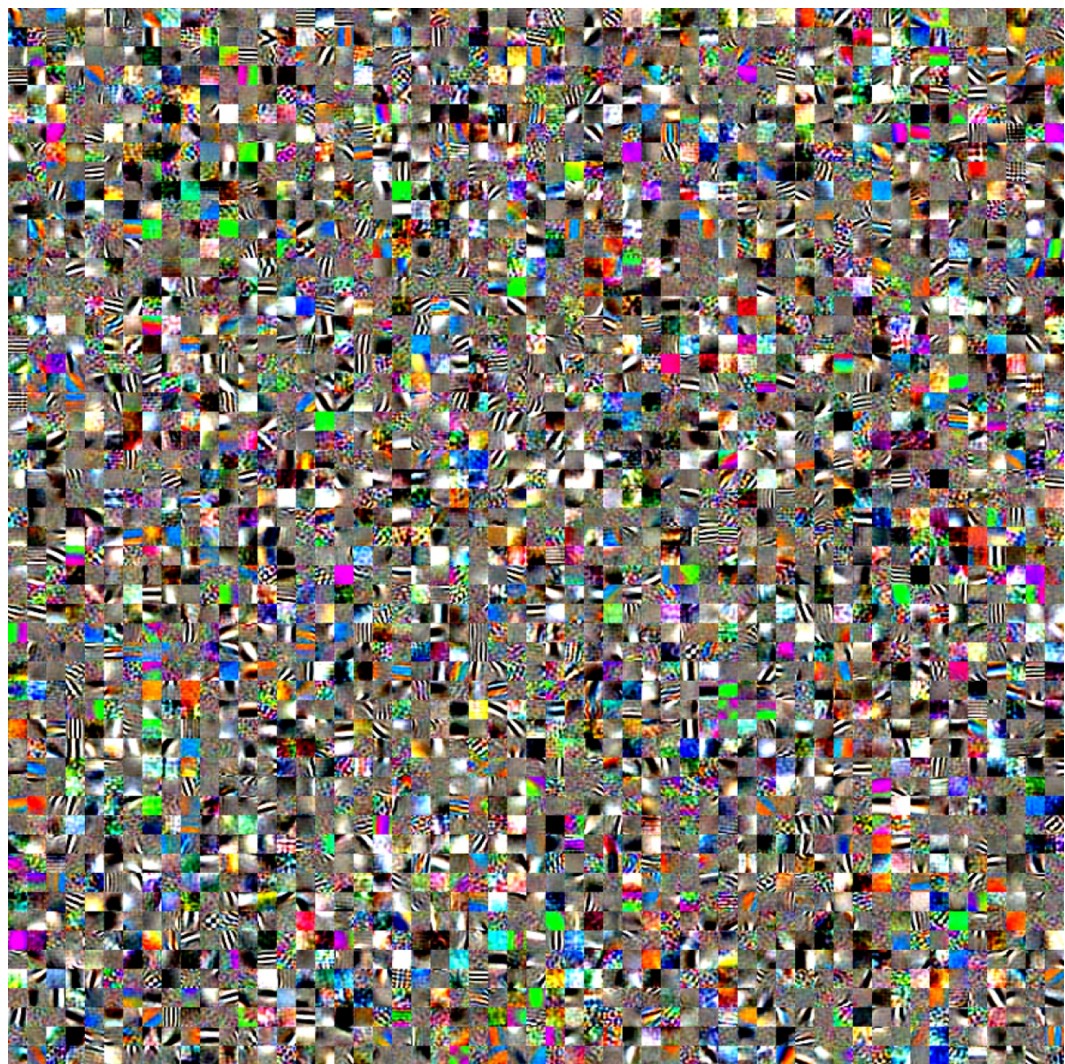

Figure 5: Visualizing a randomly selected 3025 patch memories of the 3072 learned by weight matrix in the Hopfield Network module (HN) of our model. These memories are vectors of the same dimensions $D$ as the patch tokens, stored as rows in the weight matrix $\xi$. Each image patch is visualized using the model's trained decoder.

### A.3 OBSERVING ENERGY DYNAMICS

We include as part of our supplemental submission a video showing the dynamics of one of our trained models through time together with the corresponding energy at every step. From the video, it is clear that the image progressively improves in quality as the energy decreases up to the point when the token representations are passed to the decoder. At the same time, we found it challenging to find the time constants and number of training steps so that the energy of each image reaches the energy minimum (fixed point) when the loss is computed. For instance, for the image shown in the video, its quality actually starts to degrade when the dynamics is allowed to run for longer than what was used at training (while the energy is still decreasing). Additionally, when training models at greater depth the gradients can vanish since many recurrent applications of the ET block are necessary. We hope to comprehensively investigate these questions/limitations in future work. We include a screenshot of the video in Figure 8 and encourage readers to watch the full video for the dynamics.

$$W_\text{keys} \qquad\qquad W_\text{queries}$$

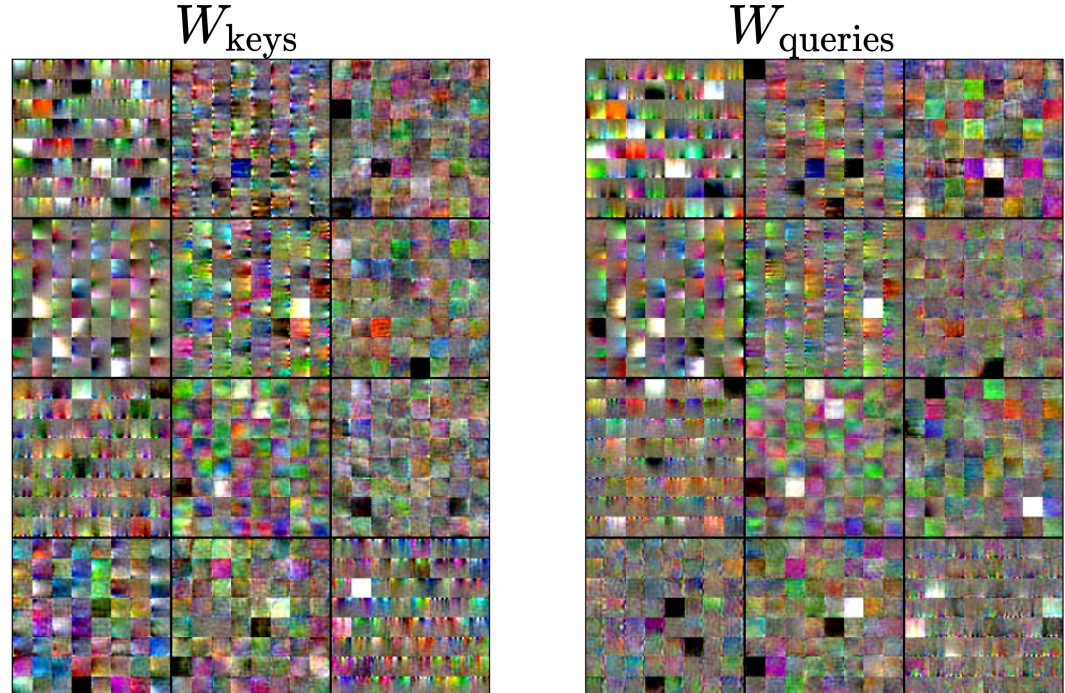

Figure 6: Visualizing the token dimension of the "key" and "query" matrices of the attention as image patches. Each head is represented as a cell on the $4 \times 3$ grid above. We use the trained decoder of our model to visualize each weight.

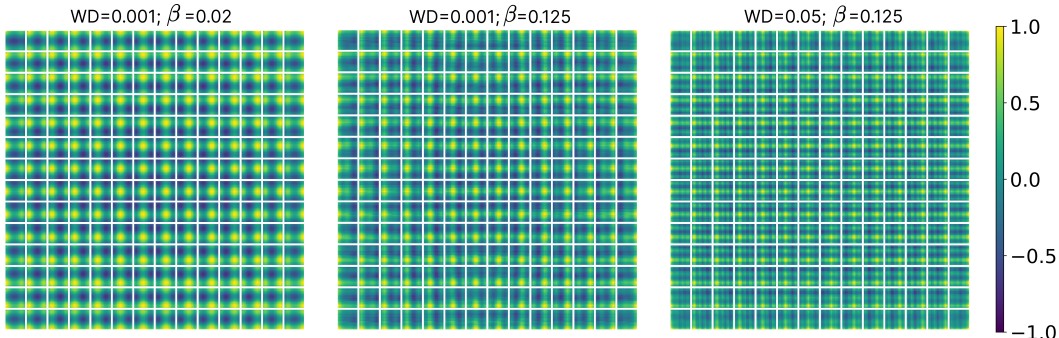

Figure 7: The cosine similarity between position biases of patches when the ET-base model is trained under different hyperparameter choices for $\beta$ (inverse temperature of the attention energy) and weight decay. Our ET sees a trend where smoother correlations are observed with smaller $\beta$ and weight decay.

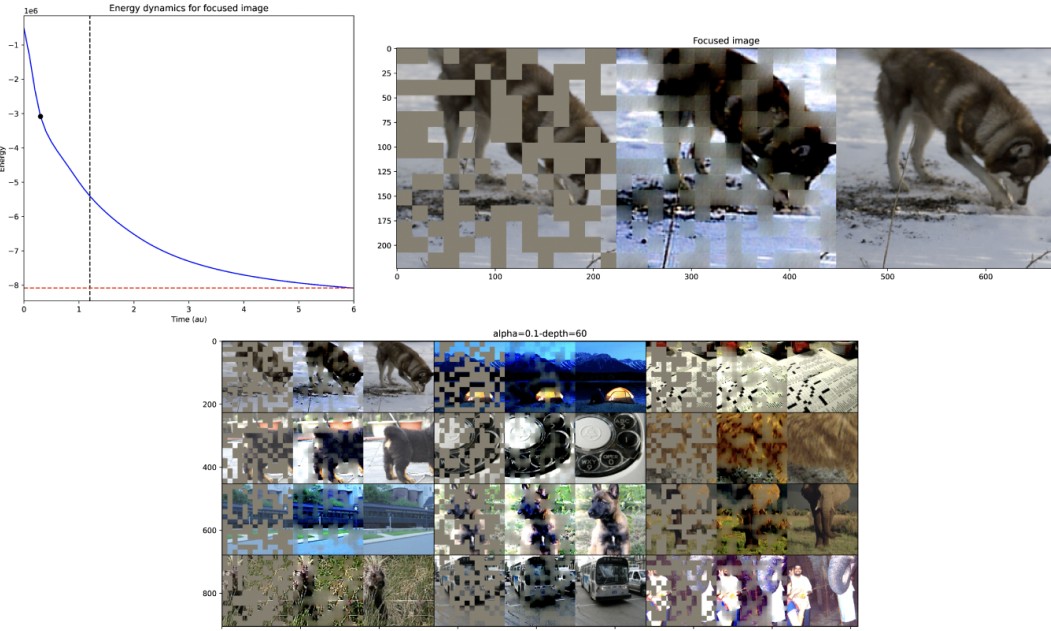

Figure 8: Screenshot of the accompanying video showcasing the energy dynamics of our model. *Top left*: the energy of our model over time (with units $\tau$) for the dog image highlighted at the *top right*. Each cell of the dog represents the (masked input, reconstructed image at time $t$, original image). We step through time using a time step of $dt = 0.1$ and record the total energy of our system on the image tokens as the black dot descending the blue energy curve. The dashed vertical black line shows the point in the energy curve where the representations were passed to the loss function when training the model, whereas the horizontal red dashed line shows the "fixed point" at the end of the simulated dynamics (in reality, the energy still descends slightly after that). *Bottom:* We display 11 other images as (masked image, reconstructed image at time $t$, original image) aligned with the time step. Each image's energy trajectory will be slightly different (not shown).

## B    DETAILS OF ET TRAINING ON ANOMALY DETECTION TASK

Graph anomaly detection refers to the process of detecting outliers that deviate significantly from the majority of the samples. Neural network based methods are very popular due to their capability of learning sophisticated data representations. DOMINANT (Ding et al., 2019) utilizes an auto-encoder framework, using a GCN as an encoder and two decoders for structural reconstruction and attribute reconstruction. ALARM (Peng et al., 2020) aggregates the encoder information from multiple view of the node attributes. Another study Zhao et al. (2020), propose a novel loss function to train graph neural networks for anomaly-detectable node representations. In Ding et al. (2021) generative adversarial learning is used to detect anomaly nodes where a novel layer is designed to learn the anomaly-aware node representation. Recently, Tang et al. (2022) pointed out that anomalies can lead to the "rightshift" of the spectral energy distribution – the spectral energy concentrates more on the high frequencies. They designed a filter that can better handle this phenomenon. We propose a new anomaly detection model from the perspective of Associative Memory (pattern matching), which does not have the over-smoothing problem often faced by GCNs, and has better model interpretability (outliers should be far from the common pattern).

### B.1    DETAILED MODEL STRUCTURE FOR THE GRAPH ANOMALY DETECTION

First, we compute the features that are passed to our energy-based transformer. Each node's features $\mathbf{y}_A \in R^F$ are mapped into the token space $\mathbf{x}_A \in R^D$, using a linear projection $\mathbf{E}$. Learnable positional embeddings $\lambda_A$ are added to this token at $t = 1$,

$$\mathbf{x}_A^{t=1} = \mathbf{E}\mathbf{y}_A + \lambda_A \tag{13}$$

At each time step the input to the ET-block is layer normalized:

$$\mathbf{g}_A^t = \text{LayerNorm}(\mathbf{x}_A^t) \tag{14}$$

Let $\mathbf{W}^Q \in R^{Y \times H \times D}$ and $\mathbf{W}^K \in R^{Y \times H \times D}$ be the query and key weight matrices, respectively. Here $Y$ is the projection dimension in the attention operation, $H$ is the number of heads. We define

$$\begin{aligned} K_{\alpha hB} &= \sum_j W_{\alpha hj}^K \, g_{jB} \\ Q_{\alpha hC} &= \sum_j W_{\alpha hj}^Q \, g_{jC} \end{aligned} \tag{15}$$

If we let $h$ indicate the index of the head, we have

$$\Delta x_{iA}^t = \sum_{C \in \mathcal{N}_A} \sum_{h,\alpha} \left[ W_{\alpha hi}^Q \, K_{\alpha hC} \, \omega_{CA} + W_{\alpha hi}^K \, Q_{\alpha hC} \, \omega_{AC} \right] + \sum_\mu \xi_{\mu i} \, r\left( \sum_j \xi_{\mu j} g_{jA} \right) \tag{16}$$

where

$$\omega_{CA} = \underset{C}{\text{softmax}}\left( \beta \sum_\gamma K_{\gamma hC} \, Q_{\gamma hA} \right) \tag{17}$$

Here $\beta$ controls the temperature of the softmax, $\mathcal{N}_A$ stands for the neighbors of node $A$ —a set of all the nodes connected to node $A$, $r$ is the ReLU function. Restriction of the attention operation to the neighborhood of a given node is similar to that used in the Graph Attention Networks (GAT), see (Veličković et al., 2017). Finally, we have residual connection

$$\mathbf{x}_A^{t+1} = \mathbf{x}_A^t + \Delta \mathbf{x}_A^t \tag{18}$$

Intuitively, the first term considers the influence (attention score) of the neighbor nodes with respect to the target node, the second term considers the influence of the target node with respect to each of its neighbor, and the third term is the contribution of the Hopfield Network module. It can be shown that the forward pass of our energy-based transformer layer minimizes the following energy function:

$$E = -\frac{1}{\beta} \sum_C \sum_h \log\left( \sum_{B \in \mathcal{N}_C} \exp\left( \beta \sum_\alpha K_{\alpha hB} \, Q_{\alpha hC} \right) \right) - \frac{1}{2} \sum_{C,\mu} r\left( \sum_j \chi_{\mu j} \, g_{jC} \right)^2 \tag{19}$$

This energy function will decrease as the forward pass progresses until it reaches a local minimum.

After $T$ iterations when the retrieval is stable, we have the final representation for each node $\mathbf{g}_A^{\text{final}}$ as

$$\mathbf{g}_A^{\text{final}} = \mathbf{g}_A^{t=1} \,||\, \mathbf{g}_A^{t=T} \tag{20}$$

where $||$ is the concatenation sign. Following Tang et al. (2022), we treat anomaly detection as semi-supervised learning task in this work. The node representation $\mathbf{g}_A^{\text{final}}$ is fed to another MLP with the sigmoid function to compute the abnormal probability $p_A$, weighted log-likelihood is then used to train the network. The loss function is as follow:

$$\text{Loss} = \sum_A \left[ \sigma\, l_A \log(p_A) + (1 - l_A) \log(1 - p_A) \right] \tag{21}$$

where $\sigma$ is the ratio of normal labels ($l_A = 0$) to anomaly labels ($l_A = 1$).

## B.2 EXPERIMENTAL DETAILS

We train all models for 100 epochs using the Adam optimizer with a learning rate of 0.001, and use the model with the best Macro-F1 on the validation set for reporting the final results on the test set. Following Tang et al. (2022), we use training ratios 1% and 40% respectively (randomly select 1% and 40% nodes of the dataset to train the model, and use the remaining nodes for the validation and testing). These remaining nodes are split 1:2 for validation:testing. The statistics of the datasets are listed in Table 3. For the four datasets used in the experiments, Amazon and Yelp datasets can be obtained from the DGL library, T-Finance and T-Social can be obtained from Tang et al. (2022).

| Dataset | $|V|$ | $|E|$ | Anomaly(%) | Features |
|---------|-------|-------|------------|----------|
| Amazon | 11944 | 4398392 | 6.87% | 25 |
| Yelp | 45954 | 3846979 | 14.53% | 32 |
| T-Finance | 39357 | 21222543 | 4.58% | 10 |
| T-Social | 5781065 | 73105508 | 3.01% | 10 |

Table 3: Summary of all the datasets.

We report the average performance of 5 runs on the test datasets. The hyperparameters of our model are tuned based on the validation set, selecting the best parameters within 100 epochs. To speedup the training process, for the large graph datasets T-Finance and T-Social, we sample a different subgraph to train for each epoch (subgraphs have $5\%$ of the nodes with respect to the whole training data). The hyperparameters include the number of hidden dimensions in ET-attention $Y$, the number of neurons $K$ in the hidden layer within the Hopfield Network Module, the number of time iterations $T$, and the number of heads $H$. The weights are learned via backpropagation, which includes embedding projection $\mathbf{E}$, positional embedding $\lambda_A$, softmax inverse temperature parameter $\beta$, ET-attention weight tensors $\mathbf{W}^Q$ and $\mathbf{W}^K$. The optimal hyperparameters used in Table 1 are reported in Table 4. The last row in that table summarizes the range of the hyperparameter search that was performed in our experiments. In general, we have observed that for small datasets (Yelp, Amazon, T-Finance) a 1 or 2 applications of our network is sufficient for achieving strong results, for larger datasets (T-Social) more iterations (3) are necessary. For even bigger dataset (ImageNet) our network needs about 12 iterations.

| Dataset | $Y$ | $K$ | $T$ | $H$ |
|---------|-----|-----|-----|-----|
| Amazon (40%) | 128 | 640 | 1 | 2 |
| Amazon (1%) | 64 | 128 | 1 | 1 |
| Yelp (40%) | 128 | 256 | 1 | 1 |
| Yelp (1%) | 128 | 256 | 1 | 1 |
| T-Finance (40%) | 128 | 256 | 1 | 3 |
| T-Finance (1%) | 128 | 256 | 1 | 1 |
| T-Social (40%) | 128 | 256 | 3 | 3 |
| T-Social (1%) | 128 | 256 | 3 | 3 |
| Range of hyperparameters | {64, 128, 256} | {2Y, 3Y, 4Y, 5Y} | {1,2,3} | {1,2,3} |

Table 4: Hyperparameters choice of our method on all the datasets.

## C  NOTATIONS

Table 5 lists all the notations used in this paper.

Table 5: Notations used in the paper.

| Notation | Description |
|---|---|
| $F$ | dimension of node's feature space |
| $D$ | dimension of token space |
| $N$ | number of tokens |
| $Y$ | number of hidden dimensions in the attention |
| $M$ | number of hidden dimensions in the Hopfield Network |
| $T$ | number of recurrent time steps |
| $H$ | number of heads |
| $k_h$ | height of each image patch |
| $k_w$ | width of each image patch |
| $P$ | number of pixels per image patch ($3 \times k_h \times k_w$) |
| $\mathbf{y}_A$ | input feature vector of node $A$ |
| $\mathbf{x}_A$ | vector representation of token $A$ |
| $x_{iA}$ | each element of vector representation of token $A$ |
| $\mathbf{g}_A$ | vector representation of token $A$ after layernorm |
| $g_{iA}$ | each element of vector representation of token $A$ after layernorm |
| $\mathbf{K}$ | key tensor |
| $\mathbf{Q}$ | query tensor |
| $K_{\alpha hB}$ | each element of the key tensor $\mathbf{K}$ |
| $Q_{\alpha hC}$ | each element of the query tensor $\mathbf{Q}$ |
| $l_A$ | label of node A on graph |

## D  ABLATION STUDY FOR ATTENTION AND HOPFIELD NETWORK MODULES

As we described in the main text the the ET network consists of two modules processing the tokens in parallel: the attention module (ATT) and the Hopfield Network module (HN). The ATT module is responsible for routing the information between the tokens, while the HN module is responsible for reinforcing the token representation to be consistent with the general expectation about the particular data domain. It is interesting to explore the contribution that these two subnetworks produce on the task performed by the network. In this section we ablate the ET architecture by dropping each of the two subnetworks and measuring the impact of the ablation on the performance.

### D.1  ON GRAPHS

The results on graphs are reported in Table 6. From this table it is clear that most of the computation is performed by the ATT block on this task, which pools the information about other tokens to the token of interest. When ATT block is kept, but HN block is removed the network looses $1\%$ or less relative to the full ET (occasional improvements of the ablated model compared to the full ET are within the statistical error bars). In contrast, removing ATT module and keeping only the HN, the ET network effectively turns into an MLP with shared weights that is recurrently applied. In this regime the network can only use the features of a given node for that node's anomalous status prediction. This results in a more significant drop in performance, which is about $5\%$ on average.

### D.2  ON IMAGES

The ablation results for image reconstruction are shown in Table 7. Each experiment was trained using the same hyperparameter settings as shown in Table 2. After training the model on IN1K, we calculate the average MSE on the reconstructed masked tokens for the validation set (using the same 50% masking ratio used for training) across 10 different random seeds for the masking.

We make several conclusions from these ablation studies.

Table 6: Ablation study with respect to ATT block and HN Block. Best results are in **bold**.

| | Datasets | Split | ATT✓\|HN✗ | ATT✗\|HN✓ | full model (Ours) |
|---|---|---|---|---|---|
| **Macro-F1** | Yelp | 1% | $62.5_{\pm 0.3}$ ▼ | $57.4_{\pm 0.5}$ ▼(-5.6) | $\mathbf{63.0_{\pm 0.6}}$ |
| | | 40% | $70.6_{\pm 0.5}$ ▼ | $71.2_{\pm 0.7}$ ▼ | $\mathbf{71.5_{\pm 0.1}}$ |
| | Amazon | 1% | $\mathbf{89.5_{\pm 0.9}}$ ▲ | $87.4_{\pm 1.0}$ ▼ | $89.3_{\pm 0.7}$ |
| | | 40% | $91.7_{\pm 0.5}$ ▼(-1.1) | $88.7_{\pm 0.3}$ ▼(-4.1) | $\mathbf{92.8_{\pm 0.3}}$ |
| | T-Finance | 1% | $84.7_{\pm 1.0}$ ▼ | $80.3_{\pm 0.6}$ ▼(-4.8) | $\mathbf{85.1_{\pm 1.0}}$ |
| | | 40% | $87.4_{\pm 0.7}$ ▼ | $82.3_{\pm 0.8}$ ▼(-5.9) | $\mathbf{88.2_{\pm 1.0}}$ |
| | T-Social | 1% | $\mathbf{79.8_{\pm 0.6}}$ ▲ | $72.7_{\pm 1.0}$ ▼(-6.4) | $79.1_{\pm 0.7}$ |
| | | 40% | $82.9_{\pm 1.0}$ ▼ | $78.6_{\pm 1.2}$ ▼(-4.9) | $\mathbf{83.5_{\pm 0.4}}$ |
| **AUC** | Yelp | 1% | $72.9_{\pm 0.3}$ ▼ | $67.4_{\pm 0.7}$ ▼(-5.8) | $\mathbf{73.2_{\pm 0.8}}$ |
| | | 40% | $83.5_{\pm 0.4}$ ▼(-1.4) | $83.1_{\pm 0.6}$ ▼(-1.8) | $\mathbf{84.9_{\pm 0.3}}$ |
| | Amazon | 1% | $90.7_{\pm 0.8}$ ▼ | $89.8_{\pm 1.2}$ ▼ | $\mathbf{91.9_{\pm 1.0}}$ |
| | | 40% | $96.8_{\pm 0.6}$ ▼ | $95.7_{\pm 0.5}$ ▼(-1.6) | $\mathbf{97.3_{\pm 0.4}}$ |
| | T-Finance | 1% | $91.7_{\pm 1.2}$ ▼ | $90.2_{\pm 0.8}$ ▼(-2.6) | $\mathbf{92.8_{\pm 1.1}}$ |
| | | 40% | $94.3_{\pm 2.6}$ ▼ | $90.2_{\pm 2.1}$ ▼ | $\mathbf{95.0_{\pm 3.0}}$ |
| | T-Social | 1% | $\mathbf{92.2_{\pm 0.8}}$ ▲ | $86.4_{\pm 0.7}$ ▼(-5.5) | $91.9_{\pm 0.6}$ |
| | | 40% | $93.1_{\pm 0.8}$ ▼ | $88.3_{\pm 1.3}$ ▼(-5.6) | $\mathbf{93.9_{\pm 0.2}}$ |

- We gain several insights regarding the use of "self-attention" in our ET (when a token patch query is allowed to consider itself as a key in the attention weights). When both self-attention and HN are present (ET-Full+Self), there is no noticeable benefit over ET-Full for a token to attend to itself. In fact, preventing the ATTN energy module from attending to itself slightly improves the performance. However, when the HN is removed (ET-NoHN*), we notice that allowing self-attention (ET-NoHN+Self) outperforms the version that prevents self-attention (ET-NoHN).

- On its own, allowing self-attention (ET-NoHN+Self) in the ATTN module performs nearly as well as the full ET at a fraction of the total parameters. However, MSE is a forgiving metric for blurry reconstructions. While ATTN can capture the global structure of the image quite well, it does so at the expense of image sharpness (Figure 4).

- As expected, removal of the ATTN energy module performs the worst, because the HN operates tokenwise and has no way to aggregate token information across the global image without ATTN.

Figure 9 shows our best performing model (ET-Full) on the qualitative image reconstructions corresponding to the *largest* errors across IN1K validation images, averaged across all masking seeds. Likewise, Figure 10 shows the *lowest* errors across IN1K validation images and masking seeds. In general, image reconstructions that require ET to produce sharp, high frequency, and high contrast lines negatively impact MSE performance.

Table 7: Module ablation tests for image reconstruction task, reporting average IN1K validation MSE on masked tokens after 100 epochs. Reported number of parameters excludes the constant number of parameters in the affine encoder and decoder.

| Model | Has ATTN? | Allow self-attn? | Has HN? | NParams (in ET block) | MSE |
|---|---|---|---|---|---|
| ET-Full **(Ours)** | ✓ | ✗ | ✓ | 3.7M | $\mathbf{0.306_{\pm 0.10}}$ |
| ET-Full+Self | ✓ | ✓ | ✓ | 3.7M | $0.312_{\pm 0.10}$ |
| ET-NoHN+Self | ✓ | ✓ | ✗ | 1.3M | $0.343_{\pm 0.10}$ |
| ET-NoHN | ✓ | ✗ | ✗ | 1.3M | $0.403_{\pm 0.11}$ |
| ET-NoATT | ✗ | ✗ | ✓ | 2.5M | $0.825_{\pm 0.20}$ |

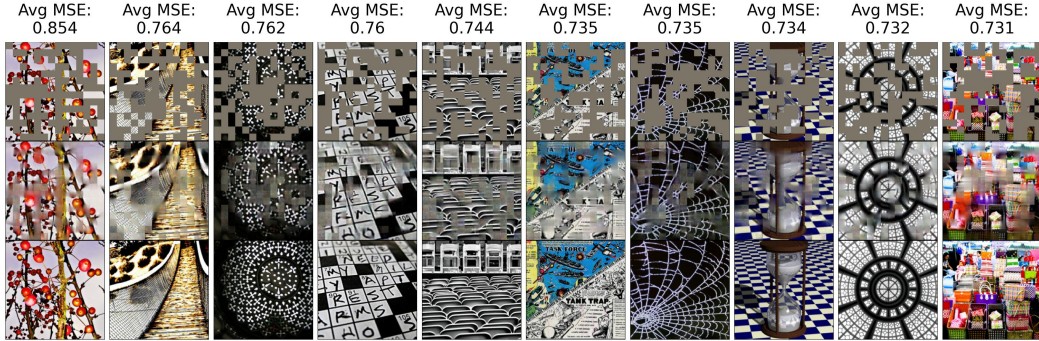

Figure 9: Reconstruction examples of images with the worst MSE from the IN1k validation set. *Top row:* input images where 50% of the patches are masked with the learned MASK token. *Middle row*: all tokens reconstructed after 12 time steps. *Bottom row*: original images.

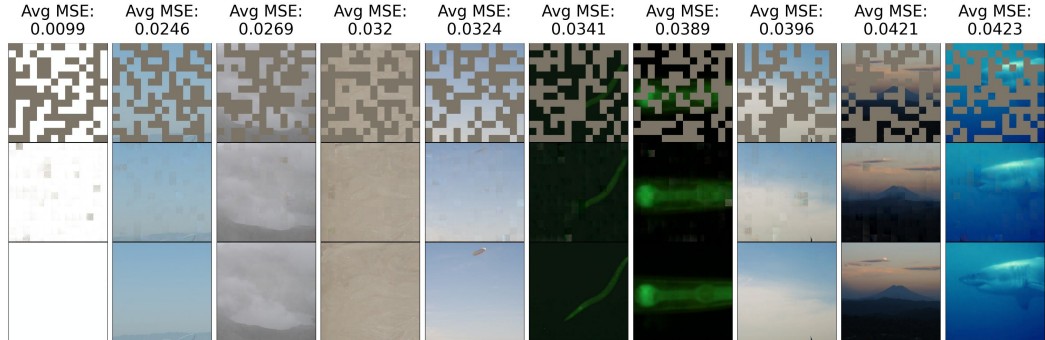

Figure 10: Reconstruction examples of images with the best (lowest) MSE from the IN1k validation set. *Top row:* input images where 50% of the patches are masked with the learned MASK token. *Middle row*: all tokens reconstructed after 12 time steps. *Bottom row*: original images.

## E  ET FOR HETEROGENEOUS GRAPH

In this section, we show the performance of our ET model in the heterogeneous graph case. For the heterogeneous graph case (where more than one type of edges exist in the graph), we first run our ET model for different subgraphs (corresponding to different types of edges), and then aggregate the final representations using two methods. We have tried max pooling and concatenation for the aggregation step. Max pooling means to pick the largest value across all the subgraph representations, and concatenation means to concatenate the representations obtained from different subgraphs. Table 8 shows the comparison between these two variants of our model and BWGNN (heterogeous case). ET performs better than heterogeneous BWGNN on Amazon, but loses to heterogeneous BWGNN on Yelp. Interestingly, BWGNN in the heterogeneous setting performs worse than BWGNN in the homogeneous setting on Amazon.

## F  GRAPH CLASSIFICATION WITH ET

As mentioned prior, GNNs have emerged as a popular approach to handle graph-related tasks due to their effective and automatic extraction of graph structural information. There have been attempts of leveraging Transformer into the graph domain, but only certain key modules, such as feature aggregation, are replaced in GNN variants by the softmax attention Ying et al. (2021). However, the Transformer model has yet to achieved competitive performance on popular leader boards of graph-level prediction compared to mainstream GNN variants Ying et al. (2021). In this section we explain how ET can be used for graph classification.

Table 8: ET for anomaly detection in heterogeneous graph setting. Best results are in **bold**.

| | Datasets | Split | MaxPool | Concatenation | BWGNN (Heterogenous) |
|---|---|---|---|---|---|
| **Macro-F1** | Yelp | 1% | $61.5_{\pm0.4}$ | $61.7_{\pm0.2}$ | **67.02** |
| | | 40% | $70.7_{\pm0.6}$ | $71.1_{\pm0.1}$ | **76.96** |
| | Amazon | 1% | $\mathbf{88.3_{\pm1.6}}$ | $87.4_{\pm1.4}$ | 83.83 |
| | | 40% | $\mathbf{92.1_{\pm0.2}}$ | $91.8_{\pm0.2}$ | 91.72 |
| **AUC** | Yelp | 1% | $72.2_{\pm0.5}$ | $72.8_{\pm0.2}$ | **76.95** |
| | | 40% | $84.3_{\pm0.4}$ | $85.1_{\pm0.1}$ | **90.54** |
| | Amazon | 1% | $\mathbf{90.8_{\pm1.4}}$ | $90.6_{\pm1.0}$ | 86.59 |
| | | 40% | $\mathbf{97.5_{\pm0.1}}$ | $97.2_{\pm0.6}$ | 97.42 |

## F.1 DETAILS OF GRAPH CLASSIFICATION ET MODEL

Given a graph $G = (V, A)$, where $V = \{v_1, v_2, \ldots, v_N\}$ and $A \in \{0,1\}^{N \times N}$ is the adjacency matrix of the graph, each feature vector $\mathbf{x_B} \in \mathbb{R}^F$ corresponding to node $v_B$ is first projected to the token space $\tilde{\mathbf{x}}_B \in \mathbb{R}^D$ via a linear embedding. Then, the CLS token $\tilde{\mathbf{x}}_{\text{CLS}}$ is concatenated to the set of tokens resulting in $\tilde{X} \in \mathbb{R}^{(N+1) \times D}$ and the positional embedding $\tilde{\lambda} \in \mathbb{R}^{(N+1) \times D}$ is added afterwards.

To obtain the positional embedding $\tilde{\lambda}$, the adjacency matrix $A$ is first padded in the upper left corner with ones resulting in $\tilde{A} \in \{0,1\}^{(N+1) \times (N+1)}$. This particular step provides the CLS token full connectivity with all of the nodes in the graph. The top $k$ smallest eigen-vectors $\lambda \in \mathbb{R}^{(N+1) \times k}$ are then obtained from the eigen-value decomposition of the unnormalized Laplacian matrix $\tilde{L}$

$$\tilde{L} = \tilde{D}^{-\frac{1}{2}} \tilde{A} \tilde{D}^{-\frac{1}{2}} \tag{22}$$

and projected to the token space via a linear embedding to form the positional embedding $\tilde{\lambda} \in \mathbb{R}^{(N+1) \times D}$.

Meanwhile, the attention in ET is modified to take in $\hat{\mathbf{A}} \in \mathbb{R}^{H \times (N+1) \times (N+1)}$ the parameterized adjacency tensor, which acts as the weighted 'attention mask' that enables the model to consider the graph structural information. To obtain $\hat{\mathbf{A}}$, a 2D-convolutional layer with $H$ filters equals to the number of heads in the attention block, 'SAME' padding, and a stride of 1 is performed on the outer product of $\tilde{X}$ to itself. The result is then multiplied with $\tilde{A}$ element-wise (denoted by $\odot$) via broadcasting.

$$\hat{\mathbf{A}} = \text{Conv2D}(\tilde{X} \otimes \tilde{X}) \odot \tilde{A} \tag{23}$$

Altogether, the resulting energy equation is

$$E^{\text{ATT}} = -\frac{1}{\beta} \sum_h \sum_C \log \left( \sum_{B \neq C} \exp \left( \beta \sum_\alpha K_{\alpha h B} \, Q_{\alpha h C} \odot \hat{A}_{\alpha h C} \right) \right) \tag{24}$$

Additionally, in this implementation, the overall model is consisted of $S$ vertically stacked ET blocks, where each block shares the same number of $T$ depth and has a different LayerNorm. Similarly, the token representation $\tilde{X}^{t,\ell}$ at each dynamic step $t$ corresponding to a block $\ell$ is first layer-normalized. Keep in mind, $\tilde{X} = \tilde{X}^{t=1, \ell=1}$ is the initial token representation.

$$\mathbf{g}^{t,\ell} = \text{LayerNorm}(\tilde{X}^{t,\ell}) \tag{25}$$

Following the dynamic equations 6, we inject a small amount of noise $\epsilon^{t,\ell} \in (0,1)$, generated from a normal distribution with standard deviation of 0.02 and zero mean, into the gradient of energy function to produce $\tilde{X}^{t+1,\ell}$, the new token representation of block $\ell$. The premise of this noise injection is to 'robustify' the model and help push it towards a local minimum of the energy function.

$$\tilde{X}^{t+1,\,\ell} = \tilde{X}^{t,\,\ell} - \alpha(\nabla_{\mathbf{g}} E^{t,\,\ell} + \epsilon^{t,\,\ell}) \tag{26}$$

Once stability is reached in the retrieval dynamics of block $\ell$, the final representation $X^{t=T,\,\ell}$ is then passed on to the next block $\ell + 1$ and the whole process is repeated again. When the final token representation $\hat{Y} = \tilde{X}^{t=T,\,\ell=S}$ is computed by the last block $S$, the resultant CLS token $\hat{Y}_0 \in \mathbb{R}^{D'}$ extracted from $\hat{Y}$ is utilized as the predictor of the current graph $G$.

## F.2 Experimental Evaluation

Eight datasets of the TUDataset Morris et al. (2020) collection are used for experimentation. NCI1, NCI109, MUTAG, MUTAGENICITY, and FRANKENSTEIN are a common class of graph datasets consists of small molecules with class labels representing toxicity or biological activity determined in drug discovery projects Morris et al. (2020). Meanwhile, DD, ENZYMES, and PROTEINS represent macromolecules. The task for both DD and PROTEINS is to classify whether a protein is an enzyme. Lastly, for ENZYMES, the task is to assign enzymes to one of the 6 EC-top-level classes, which reflect the catalyzed chemical reaction Morris et al. (2020).

We compare ET with the current state-of-the-art approaches for the mentioned datasets, which include WKPI-kmeans Zhao & Wang (2019), WKPI-kcenters Zhao & Wang (2019), DSGCN Balcilar et al. (2020), HGP-SL Zhang et al. (2021), U2GNN Nguyen et al. (2022), and Evolution Graph Classifier (EvoG) Domingue et al. (2019). Additionally, approaches that are close to the baselines are included to further contrast the performance of our model. Following the 10-fold cross validation process delineated in Morris et al. (2020), accuracy score is used as the evaluation metric and reported in Table 9.

Table 9: Performance of all the methods on the graph classification datasets, where additional features are used if exist. Following Morris et al. (2020), mean and standard deviation obtained from 10 runs of 10-fold cross validation are reported and the baselines (standard deviations are only included if they are available in the prior work). If the entry is unavailable in prior literature it is denoted by '-'; best results are in **bold**. The performance difference between non-baseline approaches (including ours) and the baseline (specified by the gray cell in each column) is indicated by ▼(decrease) and ▲(increase) along with the value.

| Method | PROTEINS | NCI1 | NCI109 | DD | ENZYMES | MUTAG | MUTAGENICITY | FRANKENSTEIN |
|---|---|---|---|---|---|---|---|---|
| WKPI (kmeans) | $78.5_{\pm0.4}$ ▼(6.4) | **$87.5_{\pm0.5}$** | $85.9_{\pm0.4}$ ▼(1.5) | $82.0_{\pm0.5}$ ▼(13.7) | - | $85.8_{\pm2.5}$ ▼(14.2) | - | - |
| WKPI (kcenters) | $75.2_{\pm0.4}$ ▼(9.7) | $84.5_{\pm0.5}$ ▼(3.0) | **$87.4_{\pm0.3}$** | $80.3_{\pm0.4}$ ▼(15.4) | - | $88.3_{\pm2.6}$ ▼(11.7) | - | - |
| Spec-GN | - | $84.8_{\pm1.6}$ ▼(2.7) | $83.6_{\pm0.8}$ ▼(3.8) | - | $72.5_{\pm5.8}$ ▼(5.9) | - | - | - |
| Norm-GN | - | $84.9_{\pm1.7}$ ▼(2.6) | $83.5_{\pm1.3}$ ▼(3.9) | - | $73.3_{\pm8.0}$ ▼(5.1) | - | - | - |
| GWL-WL | $75.8_{\pm0.6}$ ▼(9.1) | - | - | - | $71.3_{\pm1.1}$ ▼(7.1) | - | - | $78.9_{\pm0.3}$ |
| HGP-SL | $84.9_{\pm1.6}$ | $78.5_{\pm0.8}$ ▼(9.1) | $80.7_{\pm1.2}$ ▼(6.7) | $81.0_{\pm1.3}$ ▼(14.7) | $68.8_{\pm2.1}$ ▼(9.6) | - | $82.2_{\pm0.6}$ | - |
| DSGCN | $77.3_{\pm0.4}$ ▼(7.6) | - | - | - | $78.4_{\pm0.6}$ | - | - | - |
| U2GNN | $80.0_{\pm3.2}$ ▼(4.9) | - | - | $95.7_{\pm1.9}$ | - | $88.5_{\pm7.1}$ ▼(11.5) | - | - |
| NDP | $73.4_{\pm3.1}$ ▼(11.5) | $74.2_{\pm1.7}$ ▼(13.3) | - | $72.8_{\pm5.4}$ ▼(22.9) | $44.5_{\pm7.4}$ ▼(34.9) | $87.9_{\pm5.7}$ ▼(12.1) | $77.9_{\pm1.4}$ ▼(4.3) | - |
| ASAP | $74.2_{\pm0.8}$ ▼(10.7) | $71.5_{\pm0.4}$ ▼(16.0) | $70.1_{\pm0.6}$ ▼(17.3) | $76.9_{\pm0.7}$ ▼(18.8) | - | - | - | $66.3_{\pm0.5}$ ▼(12.6) |
| EvoG | - | - | - | - | $55.7$ ▼(22.7) | $100.0$ | - | - |
| ET (Ours) | $78.9_{\pm0.9}$ ▼(6.0) | $83.6_{\pm0.2}$ ▼(4.0) | $82.4_{\pm0.2}$ ▼(5.0) | $84.6_{\pm0.3}$ ▼(11.1) | $93.8_{\pm0.4}$ ▲15.4 | $99.7$ ▼(0.3) | **$88.3_{\pm0.2}$** ▲6.2 | **$99.9_{\pm0.03}$** ▲21.0 |

In general, we have observed that the modified ET demonstrates consistent performance across all datasets that is near the current state-of-the-art approaches. Based on the statistics of the experimental datasets in table 10, ET performs extremely well when trained on large graph datasets (e.g., MUTAGENICITY and FRANKENSTEIN). However, with respect to NCI1, NCI109, and DD datasets, there remains an open question on which graph characteristics (e.g., assortativity and density) impair the performance of the model.

## F.3 Experimental Details

In the graph domain, it is common to concatenate all of the feature vectors of all graphs in a batch together. However, in order for ET to work, we form the batch dimension by separating the feature vectors of all graphs

in a given batch and utilize the largest node count to pad all graphs such that they all share the same number of nodes. Additionally, we set a limit on the number of nodes (set as 400) to prevent out-of-memory error. Specifically, if a graph has a node count exceeding the limit, the number of utilized nodes is equal to the limit. Hence, a portion of the graph structural information is ignored as a result. However, it is worth mentioning such a graph is rare in the experimental datasets.

Table 10: Graph classification dataset statistics and properties (additional node attributes are indicated by '+' if exist).

| Dataset | Graphs | Avg. Nodes | Avg. Edges | Node Attr | Classes |
|---|---|---|---|---|---|
| MUTAG | 188 | 17.93 | 19.79 | 7 | 2 |
| ENZYMES | 600 | 32.63 | 62.14 | 18 + 3 | 6 |
| PROTEINS | 1113 | 39.06 | 72.82 | 0 + 4 | 2 |
| DD | 1178 | 284.32 | 715.66 | 89 | 2 |
| NCI1 | 4110 | 29.87 | 32.30 | 37 | 2 |
| NCI109 | 4127 | 29.68 | 32.13 | 38 | 2 |
| MUTAGENICITY | 4337 | 30.32 | 30.77 | 14 | 2 |
| FRANKENSTEIN | 4337 | 16.90 | 17.88 | 780 | 2 |

We train all models for 200 epochs using AdamW Loshchilov & Hutter (2017). The best model is selected based on its performance obtained from the 10-fold cross validation process delineated in Morris et al. (2020). Since the task is classification, all models are trained with the cross-entropy loss function with no temperature. Additionally, the cosine-annealing with warm-up learning rate scheduler is utilized, where the initial and peak learning rates are set as $5e - 8$ and $0.001$, respectively. The number of warm-up steps is set to 30 epochs while the batch size is 64 for all datasets with the exception of the DD dataset, which requires a batch size of 256 for faster training time. The whole experiment is implemented using JAXBradbury et al. (2018), Flax (Heek et al., 2020), Jraph Godwin* et al. (2020), and PyTorch Geometric Fey & Lenssen (2019) packages.

Lastly, we report the average performance of 10 runs on the 10-fold cross validation process with random seeding. The hyperparameters of our model are tuned based on the performance of the cross validation, selecting within 100 epochs. The optimal hyper-parameters are reported in table 11 and the statistics of the used datasets are reported in table 10.

Table 11: Hyperparameter and architecture choices for ET during graph classification training experiments.

| Training | | | Architecture | |
|---|---|---|---|---|
| batch_size | 64 | | token_dim | 128 |
| batch_size$_{DD}$ | 256 | | num_heads | 12 |
| epochs | 200 | | head_dim | 64 |
| lr | 1e-3 | | $\beta$ | $\frac{1}{\sqrt{64}}$ |
| warmup_epochs | 30 | | train_betas | No |
| start & end lr | 5e-7 | | step size $\alpha$ | 0.01 |
| b1, b2 (ADAM) | 0.9, 0.99 | | k eigenvalues | 15 |
| weight_decay | 0.05 | | depth | 2 |
| grad_clipping | None | | block_size | 2 |
| | | | kernel_size | [3, 3] |
| | | | dilation_size | [1, 1] |
| | | | hidden_dim HN | 512 |
| | | | bias in HN | None |
| | | | bias in ATT | None |
| | | | bias in LNORM | Yes |
| | | | no. of params | 530,084 |
| | | | no. of params per block | 262,929 |

## G  FORMAL ALGORITHM

We describe the algorithm for the training and inference of ET in algorithm 1, assuming backpropagation through time using SGD. Symbols not defined in the algorithm itself are reported in Table 5. We define the **Infer** function to operate independently over each item in a batch.

# H  PARAMETER COMPARISON

The energy function enforces symmetries in our model, which means ET has fewer parameters than its ViT counterparts. In particular, ET has no "Value Matrix" $\mathbf{W}^V$ in the attention mechanism, and the HN module has only one of the two matrices in the standard MLP of the traditional Transformer block. We report these differences in Table 12. We take the ET configuration used in this paper, which has an architecture fully comparable to the original ViT-base (Dosovitskiy et al., 2021) with `patch_size=16`, and report the number of parameters against ViT-base and an "ALBERT" version of ViT (Lan et al., 2020) where a single ViT block is shared across layers. We saw no benefit when including biases in ET, so we also exclude the biases from the total parameter count in the configuration of ViT and ALBERT-ViT. We report both the total number of parameters and the number of parameters per Transformer block.

Table 12: Comparison between the number of parameters in a standard ViT, an ALBERT version of ViT where standard Transformer blocks are shared across layers, and our ET. Comparison is done assuming no biases in any operation.

| Model | NParams | | NParams (per block) | |
|---|---|---|---|---|
| ViT-Base | 86.28M | ▼0.00% | 7.08M | ▼0.00% |
| ALBERT_ViT-Base | 8.41M | ▼90.25% | 7.08M | ▼0.00% |
| **ET** | **4.87M** | **▼94.36%** | **3.54M** | **▼50.02%** |

---

**Algorithm 1:** Training and inference pseudocode of ET for image reconstruction task

---

1  **HyperParameters**
2    $\alpha$: Energy descent stepsize
3    $\epsilon$: Learning rate
4    $p$: Token mask probability
5    $b$: batch size

6  **Parameters**
7    $\mathbf{W}^K \in \mathbb{R}^{Y \times H \times D}, \mathbf{W}^Q \in \mathbb{R}^{Y \times H \times D}$: Key, Query kernels of the Energy Attention
8    $\xi \in \mathbb{R}^{M \times D}$: Kernel of Hopfield Network
9    $\gamma_{\text{norm}} \in \mathbb{R}, \delta_{\text{norm}} \in \mathbb{R}^D$: Scale, bias of LayerNorm
10    $\text{MASK} \in \mathbb{R}^D$: Mask token
11    $\delta_{\text{pos}} \in \mathbb{R}^{N \times D}$: Position bias, added to each token
12    $\mathbf{W}_{\text{enc}} \in \mathbb{R}^{P \times D}, \delta_{\text{enc}} \in \mathbb{R}^D$: Kernel, bias of affine Encoder
13    $\mathbf{W}_{\text{dec}} \in \mathbb{R}^{D \times P}, \delta_{\text{dec}} \in \mathbb{R}^D$: Kernel, bias of affine Decoder

14  **Infer**
15    **Inputs**
16       Corrupted image tokens $\tilde{X} \in \mathbb{R}^{N \times D}$

17    Add position biases: $\tilde{X} \leftarrow \tilde{X} + \delta_{\text{pos}}$;
18    **for** timesteps $t = 1, \ldots, T$ **do**
19       Normalize each token:
20          $\tilde{g} \leftarrow \text{LayerNorm}(\tilde{X}; \gamma_{\text{norm}}, \delta_{\text{norm}})$;         $\tilde{g} \in \mathbb{R}^{N \times D}$
21       Calculate Energy of tokens:
22          $E \leftarrow \text{EnergyTransformer}(\tilde{g}; \mathbf{W}^K, \mathbf{W}^Q, \xi)$;      $E \in \mathbb{R}$
23       $\tilde{X} \leftarrow \tilde{X} - \alpha \nabla_{\tilde{g}} E$;
24    **return** $\tilde{X}$

25  **Train**
26    **Inputs**
27       Dataset $S_{\text{train}}$ with elements $X \in \mathbb{R}^{\text{channels} \times \text{height} \times \text{width}}$
28    **Initialize**
29       Randomly initialize from $\mathcal{N}(0, 0.02)$:
30          $\mathbf{W}^K, \mathbf{W}^Q, \xi, \text{MASK}, \mathbf{W}_{\text{enc}}, \mathbf{W}_{\text{dec}}, \delta_{\text{pos}} \sim \mathcal{N}(0, 0.02)$
31       Set other biases to zero: $\delta_{\text{enc}}, \delta_{\text{dec}}, \delta_{\text{norm}} \leftarrow 0$
32       Set LayerNorm scale to one: $\gamma_{\text{norm}} \leftarrow 1$

33    **for** epoch $n = 1, \ldots, N_{\text{epoch}}$ **do**
34       $S_{\text{epoch}} \leftarrow S_{\text{train}}$
35       **for** batch $B \subset S_{\text{epoch}}$        $B \in \mathbb{R}^{b \times \text{channels} \times \text{height} \times \text{width}}$
36       **do**
37          Convert image into non-overlapping patches:
38             $B_{\text{patch}} \leftarrow \text{Patchify}(B)$;       $B_{\text{patch}} \in \mathbb{R}^{b \times N \times P}$
39          Embed image patches into tokens:
40             $X \leftarrow \text{Encode}(B_{\text{patch}}; \mathbf{W}_{\text{enc}}, \delta_{\text{enc}})$;     $X \in \mathbb{R}^{b \times N \times D}$
41          Replace image tokens randomly by $\text{MASK}$:
42             $\tilde{X}, I_{\text{mask}} \leftarrow \text{Mask}(X; \text{MASK}, p)$     $\tilde{X} \in \mathbb{R}^{b \times N \times D}, I_{\text{mask}} \in \{0, 1\}^{b \times N}$
43          Reconstruct tokens with ET:
44             $\tilde{X} \leftarrow \textbf{Infer}(\tilde{X})$
45          Decode tokens:
46             $\hat{B}_{\text{patch}} \leftarrow \text{Decode}(\tilde{X}[I_{\text{mask}}]; \mathbf{W}_{\text{dec}}, \delta_{\text{dec}})$;   $\hat{B}_{\text{patch}} \in \mathbb{R}^{b \times N \times P}$
47          Calculate MSE loss on corrupted tokens:
48             $L \leftarrow \text{Mean}(|\hat{B}_{\text{patch}}[I_{\text{mask}}] - B_{\text{patch}}[I_{\text{mask}}]|^2)$    $L \in \mathbb{R}$
49          params $\leftarrow$ params $- \epsilon \nabla_{\text{params}} L$
50          $S_{\text{epoch}} \leftarrow S_{\text{epoch}} \setminus B$

51    **return** params

---

