# OpenReview forum: "Energy Transformer"
_ICLR.cc/2023/Conference — Submitted to ICLR 2023_

### Official Review · Reviewer_VTZR · 2022-10-24

**Confidence:** 3
**Correctness:** 3
**Technical Novelty And Significance:** 2
**Empirical Novelty And Significance:** 2
**Recommendation:** 6

**Clarity, Quality, Novelty And Reproducibility:**

Unclear descriptions reduce clarity and reproducibility.

Technical novelty should be clarified.

**Strength And Weaknesses:**

[Strengths]
+ The transformer is a prominent attention mechanism in machine learning, and it is interesting to design the Transformer mechanism from energy attention and the Hopfield network.

[Weaknesses]
1) The connection between Hopfield Networks, energy function, and Transformer is mentioned in Ramsauer et al. (2020). It is not clear the new insight of the proposed Energy Transformer.
2) The effect of designing the proposed ET block is not apparent; it is better to conduct an ablation study experiment to assess each component within such a designed block.
3) The detailed structure of the Energy Transformer is not explicitly depicted. A network architecture helps the reader to understand the ET implementation.
4) Since the Energy Transformer requires recurrently updating, it is better to show the cost of training and testing compared with the vanilla Transformer.
5) The experiments are weak. The quantitative result has merely the graph anomaly detection. However, the results of BWGNN (Hetero) and [A], which shows good performance, are not included. The on-par performance of the Energy Transformer does not make it impressive to the reader.

Related paper:
[A] Derek Lim, Xiuyu Li, Felix Hohne, and Ser-Nam Lim: New Benchmarks for Learning on Non-Homophilous Graphs. Workshop on Graph Learning Benchmarks, WWW 2021.


**Summary Of The Paper:**

This paper proposes a transformer architecture called Energy Transformer that replaces the sequence of feedforward transformer blocks with a single large Associative Memory model. The sequence of the energy transformer layers is designed to minimize an energy function in charge of representing the relationships between the tokens. Experiments show that Energy Transformer achieves good performances on image completion and graph anomaly detection.

**Summary Of The Review:**

The primary concern of this paper is its weak experiments, which do not sufficiently demonstrate the strength of the proposed Energy Transformer.

---

> ### Author Response · Authors · 2022-11-19
> **Response Reviewer VTZR [part 2 of 2]**
>
> > Since the Energy Transformer requires recurrently updating, it is better to show the cost of training and testing compared with the vanilla Transformer.
>
>
> The parameter count of the Energy Transformer compared to the standard Vision Transformer is summarized in Appendix H. The enforced symmetries of our energy function (i.e., no value matrix in the ATTN, no second kernel in the MLP) make the ET have about half the number of parameters per block compared to the standard Transformer block. The fact that ET is recurrent does not inherently increase the computational cost provided that the depth of ET is set to the same depth as the standard ViT. Thus, from the perspective of memory footprint of the model, ET is actually more efficient than conventional transformers with unshared weights. One could expect that recurrent updates may result in vanishing gradients, which may cause problems training the network using the backpropagation through time algorithm. Despite this potential problem, as evidenced by our empirical results, we were able to find the set of parameters such that the network trains well and achieves strong performance across multiple tasks.
>
> > The experiments are weak. The quantitative result has merely the graph anomaly detection. However, the results of BWGNN (Hetero) and [A], which shows good performance, are not included. The on-par performance of the Energy Transformer does not make it impressive to the reader.
>
>
> We have included two additional results:
> - Graph classification results, please see Appendix F.
> - The results for anomaly detection in heterogeneous setting are added to Appendix E.
>
> We want to reemphasize that the main contribution of this work is theoretical. The goal of empirical evaluations is to demonstrate that ET can perform well (on-par with existing techniques) on a diverse set of tasks, not to beat SOTA benchmarks.

---

> ### Author Response · Authors · 2022-11-19
> **Response Reviewer VTZR [part 1 of 2]**
>
> Thank you for your feedback. We address the questions below.
>
> > The connection between Hopfield Networks, energy function, and Transformer is mentioned in Ramsauer et al. (2020). It is not clear the new insight of the proposed Energy Transformer.
>
> You are correct that this previous paper mentions Hopfield Networks, transformers, and energy functions, however, our work makes conceptual theoretical statements far beyond the proposal of Ramsauer et al. (2020) in several aspects:
>
> - That previous work only connects the attention operation inside the transformer block with Hopfield Networks. In contrast, in our work the **entire block** (including attention, layer normalization, skip connections, feed-forward network) is conceptualized as a large Hopfield Network.
> - The correspondence between the attention operation and the Hopfield Network in Ramsauer et al. (2020) holds only if one restricts the Hopfield Network dynamics to a single step update. For this reason the energy interpretation of the attention operation in that prior work is **only approximate**, and provides limited insights about the computational strategy of the attention operation in transformers. It is impossible to claim that the outputs of the transformer correspond to low energy states in that previous approach. In contrast, our Energy Transformer minimizes the novel energy function proposed in this work in full generality -- **without any approximations**. The outputs of our ET network correspond to the low energy states in a precise mathematical sense.
> - Our network is **recurrent**. As tokens propagate through the layers the mathematical structure of the Hopfield Network pushes the tokens to the low energy configurations. The transformer discussed in Ramsauer et al. (2020) is **feed-forward**, i.e. there is no way to have a coherent energy function that describes evolution of the tokens across the layers.
> - The presence of the second term in the dynamical equations for the ET network, see page 6, which is absent in Ramsauer et al. (2020),  clearly indicates that our network is different from that prior work even at the level of the attention operation.
>
> These four differences with the prior work represent a substantial amount of theoretical research that we have done in order to enable meaningful interpretation of transformers' computation using the energy function perspective. Without this work, which is the main message of our paper, such an interpretation is mathematically impossible.
>
> > The effect of designing the proposed ET block is not apparent; it is better to conduct an ablation study experiment to assess each component within such a designed block.
>
> Thank you for raising this question. We have conducted an extensive ablation study for removing subnetwork modules and including/excluding self-attention terms in Appendix D.
>
> > The detailed structure of the Energy Transformer is not explicitly depicted. A network architecture helps the reader to understand the ET implementation.
>
> The network architecture is shown in Figure 2 (left). Additionally, we have added a formal algorithmic description of the ET network to Appendix G. We hope this addresses your concern.

---

### Official Review · Reviewer_v3Fm · 2022-10-24

**Confidence:** 4
**Clarity, Quality, Novelty And Reproducibility:** This paper is in general well written…
**Correctness:** 3
**Technical Novelty And Significance:** 3
**Empirical Novelty And Significance:** 2
**Recommendation:** 5

**Strength And Weaknesses:**

Strength:
In general, I like the idea of this paper:
1. This paper propose an specially designed energy-term that may be helpful for understanding the transformer structure.
2. It derives a new transformer structure based on this handcrafted energy-term
3. The authors show that this structure outperforms baselines in the graph anomaly detection task.

Possible concerns:
1. While the authors showcase experiments on image completion and graph anomaly detection task. I think these are not enough for me to draw the conclusion that this newly designed arctecture does have advantage over the previous one. The transformer structure is famous because it performs well in various task in computer vision(CV) and netural language processing(NLP). To really demonstrate the ability of a new structure, the authors need to show competitive results on those widely accepted tasks like transfer learning to ImageNet in CV or results on GLUE [1] benchmark in NLP. Currently, the authors only showcase two tasks that are well-suited for their model (and they only show quantative results in one of them), thus the general ability of this model is still unclear to me.

2. Some designs of the model are not fully justified experimentally. For example, unlike traditional transformer architecture which uses different weights for different layers, ET apply a single block recurrently and the MLPs in their hopfiled network also share weights . The authors may find theoretical explanation for this design according to their energy design. However, whether this helps or hurts the performance is unclear. Empirically speaking, recurrent model or model with shared weight may be hard to optimize due to the vanishing gradient problem. Thus, ablation study may be needed to justify the these design choices.

3. Questions for the graph anomaly detection task: How are the edges used by the model? Does the attention layer only compute attention over the nodes that have edges between them? Also, can the model be also used on directed graph? Or it is only limited to undirected graph?

4. For the image completion task, the authors assumes the image patches can be represented as fully masked and fully open area. But what if the masked area is not divisible by the patches? Then do you need to re-design the patch size (which may influence the full structure of the model)? This might not be a problem if one just treat image completion as a self-supervised task to learn representations and then applied these learned features to downstream tasks. But if the image completion task itself is the target and its performance is used to showcase the ability of the model. Then possible limitation may need to be considered.

[1] GLUE: A MULTI-TASK BENCHMARK AND ANALYSIS PLATFORM FOR NATURAL LANGUAGE UNDERSTANDING

=====================================================================================================

**Post rebuttal:**

For me, although in the new responses, the authors say that they are not trying to replace the original transformer architecture, what the paper mainly about is proposing a new transformer architecture based on modification of the original one. Thus, a natural question to ask is how this new design performs comparing with the original one. Although the authors show that their design principle is based on the Hopfield Networks, whether this principle does benefit the transformer architecture design needs further justification. There can be many interesting principles in modifying the designs, but whether they do make senses need to be supported by comprehensive and convincing experimental results. A valid modification to the transformer architecture needs to demonstrate its advantage over the original one. The current results, however, are far from that. The current results are mainly based in the graph domain. Solid experimental results in areas that transformers are mainly used in, like CV or NLP, are missing. I can not agree with the authors that since they expertise in graph domain, they only pick experiments in this area and miss others. As I said, if the paper narrows its objective into proposing a new technique to achieve STOA results in graph domain, then the current results are fair enough. However, if they want to make a valid and general design principle for transformer architecture, they should compare their new architecture in the areas that transformer architecture are widely used. As a reviewer, I'm not convinced by the current results and thus I still lean to reject. I would recommend the authors to polish their work with more convincing results in different areas or limit their objective to a technique that specialized in graph domain.

**Summary Of The Paper:**

In this paper, the authors propose a new transformer architecture called energy transformer (ET). ET is designed to minimize a special handcrafted energy function. This function contains two terms. The attention term aligns the keys and queries between neighbor token vectors and the hopefield network term matches the token vectors to some memory vectors. The ET layer is applied recurrently to imitate a continuous time differential equation that minized the designed energy terms. The authors try to demonstrate the ability of their model through image completion task and graph anomaly detection task.

**Summary Of The Review:**

I think this paper proposes some interesting ideas like designing a energy function and deriving a transformer structure based on this energy function. However, I think the experiments of this paper may not be strong enough to demonstrate that the new designed structure does have advantage over the original one.

---

> ### Author Response · Authors · 2022-11-19
> **Response to Reviewer v3Fm [part 2 of 2]**
>
> > Questions for the graph anomaly detection task: How are the edges used by the model? Does the attention layer only compute attention over the nodes that have edges between them? Also, can the model be also used on directed graph? Or it is only limited to undirected graph?
>
> Correct, the edges are used only for computing the attention. Attention is computed only over the set of nodes that are connected, similarly to GAT networks. This is pointed out above equation (18). Re directed graphs: this is a great question. Although we have not explored directed graphs in this work, our network can be adapted to that case. One needs to insert an additional matrix $A_{BC}$ of zeros and ones into the sum in equation (3)
>
> $$
>   E{^\text{ATT}} = -\frac{1}{\beta}\sum\limits_h\sum\limits_C \textrm{log} \left(\sum\limits_{B \neq C} A_{BC} \textrm{exp}\left(\beta \sum\limits_{\alpha} K_{ \alpha h B}  Q_{\alpha h C}\right) \right)
> $$
>
> This matrix, which is not symmetric, will indicate whether or not a node $C$ is allowed to query the node $B$. Such query is only allowed if the two nodes are connected by an edge of the approapriate direction.
>
> > For the image completion task... Then possible limitation may need to be considered.
>
> We only use the image completion task as a sandbox for illustrating the computational strategy of our theoretical proposal, not from the perspective of practical applications.

---

> > ### Comment · Reviewer_v3Fm · 2022-11-29
> > **I want to keep my original rating.**
> >
> > First, I would like to thank the authors for their responses. However, I haven't been convinced by the current results. What the paper proposes is a design principle for the transformer architecture. As I said, I think this idea makes sense. But since it is not a fully theoretical paper that uses math to strictly proves the usefulness of this principle, whether this principle does help the performance needs to be justified by sound experimental results. I don't think the current experiments is convincing enough to draw the conclusion that this new architecture beats the original transformer block design. I can understand the authors' rebuttal that "it is hard to redo all the experiments that thousands of researchers have done in a single paper", however, then they should at least do the most important ones to convince the readers. For me, the transformer architecture is so popular because their superior performance in CV or NLP tasks. In order to say that the new architecture can replace the original one, the authors need to show solid results on some of those widely accepted CV/NLP tasks. But currently, the experiments are main on graphly classification/abnormal detection (as the authors said themselves, the only CV task, image completion is just a sandbox for illustrating the model). If the target of the paper focuses on improving the performance of graph classification/abnormal detection tasks themselves, then I think the current experiments might be fair enough to justify the model. However, to draw the conclusion that the proposed principle is general enough for designing a new architecture that can replace the original transformer, I believe more experiments are needed. In summary, I think this paper has potential but is not good enough by now. Thus, I want to keep my rating as borderline reject.

---

> > > ### Author Response · Authors · 2022-11-30
> > > **A misunderstanding of the main message of our paper**
> > >
> > > We want to thank the reviewer for their comment: clearly a lot of thought went into the response, and for that we are appreciative. We are disappointed that the main message of our paper was perceived as an attempt to replace "the original transformer with the new architecture". While this might be possible in the long run after a substantial additional work by the community of researchers, this is not the claim of the current paper.
> > >
> > > Rather, we have modified the attention + MLP + skip connection + normalization steps of the conventional transformers to be compatible with the general guiding principle - the existence of the tractable energy function. This architecture has the same mathematical guarantees as the Hopfield Network. It turns out that our novel formulation demonstrates strong empirical performance on anomaly detection tasks in the graph domain. It also results in a highly interpretable architecture, which we can understand by considering the image in-painting task (on the level of image patches). This inherent interpretability of our proposed framework leads to an incredibly transparent network that a deep conventional transformer, trained on the same task, cannot replicate.
> > >
> > > We believe that the main messages of our paper are both highly novel and relevant for readers looking to theoretically reinterpret the design of modern transformer architectures. In light of this message and given the results in the paper, would the reviewer consider increasing their score?

---

> ### Author Response · Authors · 2022-11-19
> **Response to Reviewer v3Fm [part 1 of 2]**
>
> Thank you for your feedback. We are glad to hear that you like the idea of our paper. Transformers have been around for more than 5 years, and it is hard to redo all the experiments that thousands of researchers have done in a single paper. Given the expertise of our group in the graph domain we picked it for empirical evaluations. We also felt that readers would benefit from a more visual illustration of the inner-workings of our network, which convinced us to include the results on the image reconstruction task. As we tried to emphasize in the introduction the main contribution of our work is the theoretical framework for the energy-based transformer block. The goal of our paper is not to beat the benchmarks, but rather to illustrate the promise of this new idea. For this reason we leave GLUE experiments, and transfer learning to ImageNet for future work. Having said this, we understand that you want to see more results. For this reason, we have included a new benchmark - graph classification results, please see Appendix F. While performance of our model depends on the specific dataset, see Table 9, on the largest (in terms of the number of features, see Table 10) dataset -- FRANKENSTEIN -- our approach beats the previously published SOTA result by 21\%. Most importantly, ET demonstrates consistent strong results across all the datasets.
>
> > Some designs of the model are not fully justified experimentally. For example, unlike traditional transformer architecture which uses different weights for different layers, ET apply a single block recurrently and the MLPs in their hopfiled network also share weights . The authors may find theoretical explanation for this design according to their energy design. However, whether this helps or hurts the performance is unclear. Empirically speaking, recurrent model or model with shared weight may be hard to optimize due to the vanishing gradient problem. Thus, ablation study may be needed to justify the these design choices.
>
> You are correct that one could expect that the vanishing gradients problem could potentially make our model untrainable. Despite this possibility, as evidenced by our empirical results, we were able to successfully train ET on 3 different tasks (image completion, anomaly detection, graph classification) and obtain very strong results across all the benchmarks, sometimes even exceeding SOTA results. The main reason why gradients generally don't vanish (for practically relevant sets of the hyperparameters, e.g. $T=12$ steps) is that equation 6 effectively has a skip connection after discretization, $\alpha = dt/\tau$
>
> $$
> x(t+1) = x(t) - \alpha \frac{\partial E}{\partial g}
> $$
>
> This skip connection (the term $x(t)$ in the equation above) ensures that gradients smoothly propagate through the entire network.
>
> Regarding sharing the weights. This is the fundamental constraint on our model. If we don't share weights -- we don't have the energy function. Since the goal of our work is to explore the energy-based transformers, we have to share the weights. However, if one ignores this theoretical aspect, which is the central message of our paper, sharing weights is also good from a purely engineering perspective. In Table 12 we report the number of trainable parameters of our model, which is directly related to the memory footprint. ET is a much smaller model (e.g. only $6\%$ of the ViT-Base), and yet provides very decent results. Stacking multiple layers of ET (each layer has a separate set of weights) generally improves accuracy on the classification tasks. We study stacked models of ET in Appendix F, where parameter $S$ stands for the number of stacked layers.
>
> Also, please notice, that one can only benefit from the interpretability aspect of the ET model when the weights across the layers are shared. We discuss this aspect in the second paragraph on page 7, and in Figure 4.

---

### Official Review · Reviewer_UVr7 · 2022-10-24

**Confidence:** 4
**Correctness:** 4
**Technical Novelty And Significance:** 3
**Empirical Novelty And Significance:** 3
**Recommendation:** 8

**Clarity, Quality, Novelty And Reproducibility:**

The manuscript is overall quite clear and contains novel and interesting results.
The experimental settings are fully detailed and the code will be publicly released.

**Strength And Weaknesses:**


The manuscript is an enjoyable read and contains a good degree of novelty.
The output of the network is given by the final outcome of an energy minimization process.
Although this perspective is not new (see Yang et al '22), it has not been quite explored yet
and I find the particular instantiation given in the paper quite inspiring.
I think this could trigger additional theoretical and applicative results.
The resulting architecture is also quite simple and has a good degree of interpretability.
The paper is well organized, although some details given in the appendix could be moved to the main text.
The results on the node outlier detection tasks are very good, outperforming concurrent methods in
most cases.

Main comments:

- Why the authors didn't attempt to benchmark the performance on a standard image classification task?
  This should be doable by pooling and putting an head on top of the final outputs, as commonly done in ViT.

- Related to previous questions, is not clear if the ET can be considered as a drop-in replacement to self-attention
  transformer blocks for arbitrary task, or if its usage should be confined to some specific settings.
  I would like the authors to comment on the limitations of the model.

- It is not clear how important the HN component is and how would perform the ET without it.

- Could the author motivate why the attention module excludes the current the considere position from the input.
What happens if you remove the restriction from the sum?

- Did the authors explor stacking multiple ET modules in a feeforward fashion?

Minor comments:
- While it is clear in the appendix, I think also the main text should mention that the DE is discretized.

- eq. 2, ignored the \bar{x} dependence on x_i when computing the derivative. Should be fixed by inserting a factor 1 / (1 - 1/D) in the definition of L

- "is then minimized to train the whole network. Above, σ is the ratio of anomalous labels (lA = 1) to
the regular label." Maybe the other way around, that is regular labels / anomalous labels?

- The main text should mention that for the graph task a neighborhood softmax (similar to GAT) is used instead of standard attention

- The authors could consider numbering more equations for readibility.

**Summary Of The Paper:**


The paper puts forward a novel sequence-to-sequence neural architecture named Energy Transformer (ET) that builds
on recently introduced generalized Hopfield networks and their connection to transformer layers.
The architecture is composed by a single block that combines two subsystems,
defined by energy functions corresponding to two different types of generalized hopfield networks,
preceded by a layer normalization operator.
One subsystem is in charge of pooling to each position the information coming from the other
positions, while the other enforces the consistency of learned representations.
The system dynamics is given by a differential equation driven by the gradient
of the energy function with respect to the renormalized features. The total
energy is proven to decrease along the trajectory.
For practical purposes, the dynamics is discretized and the block applied recurrently.
The architecture is validated qualitatively on an image occlusion task and quantitatively
on a node anomaly detection task.

**Summary Of The Review:**

I think this is a good paper that deserves acceptance.

---

> ### Author Response · Authors · 2022-11-19
> **Response to Reviewer UVr7 [part 2 of 2]**
>
>
> > Did the authors explore stacking multiple ET modules in a feedforward fashion?
>
> From the theoretical perspective, the existence of the global energy function, which is the central message of our paper, requires additional constraints on the weights in the stacked layers, which are not easy to satisfy given the gradient decent learning of those weights. However, if one is willing to ignore the existence of the global energy function for the entire network, stacking of the ET blocks might be useful from the empirical perspective. We have investigated this possibility for graph classification task. Please see Appendix F, where index $l$ refers to the index of the stacked ET blocks, and parameter $S$ refers to the number of such blocks. The general conclusion is that stacking helps achieving higher classification accuracy, as expected.
>
> > I think also the main text should mention that the DE is discretized.
>
> Done, please see the last sentence of the paragraph above equation (7).
>
> > Eq. 2, ignored the $\bar{x}$ dependence on $x_i$ when computing the derivative. Should be fixed by inserting a factor $1 / (1 - 1/D)$ in the definition of $L$
>
> Equations (1) and (2) are correct as written. The term that you are referring to ($\frac{1}{D}$ in the middle of equation below) was included in the calculation, but, in fact, is equal to zero because of the sum over index $j$
>
> $$
> \frac{\partial}{\partial x_i} \sum\limits_j (x_j-\bar{x})^2 = 2 \sum\limits_j \Big[(x_j - \bar{x}) \big(\delta_{ij} - \frac{1}{D}\big)\Big] = 2(x_i - \bar{x}) - \frac{2}{D} \big(\sum\limits_j x_j - D\bar{x}\big) = 2(x_i - \bar{x})
> $$
>
> No need to insert any additional factors into $L$.
>
> > "Above, $\sigma$ is the ratio of anomalous labels $(l_A = 1)$ to the regular label." Maybe the other way around, that is regular labels / anomalous labels?
>
> You are correct, thanks for catching the typo! The rare class should be enhanced, not suppressed. Our code for the evaluations in Table 1 used the correct loss, but the typo appeared in the paper. Corrected now.
>
> > The main text should mention that for the graph task a neighborhood softmax (similar to GAT) is used instead of standard attention.
>
> Done, please see the comment above equation (18).
>
> > The authors could consider numbering more equations for readability.
>
> Done, almost all equations in the paper are now given numbers.

---

> ### Author Response · Authors · 2022-11-19
> **Response to Reviewer UVr7 [part 1 of 2]**
>
> Thank you for your enthusiastic evaluation, we are happy to hear that you enjoyed reading our paper and find our architecture "quite inspiring"! We address your questions/concerns below:
>
> > Why the authors didn't attempt to benchmark the performance on a  standard image classification task?
>
> Transformers have been around for more than five years and have contributed to many areas of machine learning. While we completely agree with you that image classification is a very interesting task to look at, it is hard to implement every possible experiment that have been done with transformers in a single 9 page paper (reviewer v3Fm, for example, requested GLUE results for NLP, which are also outside the scope of this submission). Given the expertise of our group in the graph domain we picked it for empirical evaluations. We also felt that readers would benefit from a more visual illustration of the inner-workings of our network, which convinced us to include the results on the image reconstruction task. As we tried to emphasize in the introduction the main contribution of our work is the theoretical framework for the energy-based transformer block. We have included graph classification results to Appendix F. This task is conceptually similar to image classification. We are planning to comprehensively investigate the image classification setting, and NLP tasks in subsequent papers. Please notice that on the most challenging dataset (Frankenstein), which has the largest number of node attributes (780, see Table 10) our network outperforms previously published SOTA by 21\%! Most importantly, ET demonstrates strong performance across all the datasets. We hope that you accept the graph classification results in place of the image classification results in combination with the main theoretical message of our work.
>
> > It is not clear if the ET can be considered as a drop-in replacement to self-attention transformer blocks for arbitrary task, or if its usage should be confined to some specific settings. I would like the authors to comment on the limitations of the model.
>
> We believe that the answer to this question is YES. We expect ET to be useful for many tasks where transformers have proven to perform well. Specific tasks may require certain modifications of the energy functions. However, we expect that the energy function should generally have two contributions: aggregation of the information from other tokens, and reinforcing the current belief about the identity of the given token. Recent theoretical advances in associative memory suggest plausible candidates for these two energy terms, which we explore in this paper.
>
> > It is not clear how important the HN component is and how would perform the ET without it.
>
> Done, we have included the ablation study for the Attention and HN blocks in Appendix D, see Tables 6 and 7. The high-level conclusion is that the Attention module is more important. This is particularly clear in the image domain (please see Table 7). For graph anomaly detection this general conclusion also holds (please see Table 6). However, the features of the individual nodes are already sufficiently predictive of the anomaly status of that node. Thus, the performance drop is smaller than for images. Having said this, removing attention still gives a bigger drop in performance than removing the HN module.
>
> > Could the author motivate why the attention module excludes the current the considere position from the input. What happens if you remove the restriction from the sum
>
> This is a great question, and, in fact, we have spent a substantial amount of time investigating this subtle difference. From the theoretical perspective it is more natural to exclude self-attention from the token to itself, since there is no need to route the information to itself. This is the rationale why we have presented our theoretical framework this way, see equation 3. From the empirical perspective, it turns out that both choices perform similarly on all the tasks that we have explored. To support this claim quantitatively we have included the results for the MSE of the reconstructed images in Table 7. Both choices (with and without self-attention) give statistically indistinguishable MSE losses in the end. The same conclusions apply to graphs (not shown in the paper).

---

### Official Review · Reviewer_zHgn · 2022-10-25

**Confidence:** 2
**Correctness:** 3
**Technical Novelty And Significance:** 3
**Empirical Novelty And Significance:** 3
**Recommendation:** 6

**Clarity, Quality, Novelty And Reproducibility:**

good


**Strength And Weaknesses:**

Strength
- Despite the great performance achieved by the Transformer model, its architecture is designed empirically. This work proposes a Transformer architecture from a purely theoretical perspective.
- This work provides a deep insight into the relationship between Transforms and associative memory models. Based on the relationship, it designs a new energy function and a corresponding Transformer architecture that minimizes the energy function.

Weakness
- To show the effectiveness of ET, it is better to evaluate it on more mainstream tasks.

**Summary Of The Paper:**

The paper proposes a new transformer architecture that replaces the sequence of transformer blocks with a single associative memory model. Specifically, the model is designed to minimize a specific energy function that represents the relations between tokens. The model is evaluated on image completion and graph anomaly detection, and experiments show encouraging results.

**Summary Of The Review:**

It is an interesting work toward understanding the Transformer models. To show its effectiveness, more experiments have to be conducted on more challenging tasks.

---

> ### Author Response · Authors · 2022-11-19
> **Response to Reviewer zHgn**
>
> Thank you for your feedback. We are happy to hear that you appreciate the insights about the relationship between transformers and associative memories. Given the expertise of our group for graphs we have mainly focused on this domain for benchmarking our new model. We have also included images, since it is easier to visualise the results. As much as we would like to benchmark our model on every possible benchmark, we want to emphasize that the main message of our work is theoretical - the mathematical formalization of the recurrent transformer based on associative memory. At the same time, the empirical results look very strong. We have included additional results for graph classification tasks in the revised paper, please see Appendix F.

---

> > ### Comment · Reviewer_zHgn · 2022-12-06
> > **Good work**
> >
> > Thank you for the reply. I am not an expert in this area. Based on my general knowledge, I think it is an interesting work and of good quality.  I would like to keep my  rating.

---

### Official Review · Reviewer_iuLg · 2022-10-29

**Confidence:** 2
**Correctness:** 2
**Technical Novelty And Significance:** 3
**Empirical Novelty And Significance:** 2
**Recommendation:** 6

**Clarity, Quality, Novelty And Reproducibility:**

Although the proposed method seems novel, the methodology of the proposed transformer architecture is not clear enough to me. The experimental results are also not very satisfying, as detailed in the previous section.

**Strength And Weaknesses:**

Strength: The proposed method is interesting in the sense that it reformulates transformer architecture in a way to combine Hopefield networks and attention networks, which are recurrently optimized using two carefully designed energy functions. The empirical results look promising.

Weakness: I am concerned about the clarity of the paper. For example, in the explanation of equation (3), "The log-sum energy function (3) is minimal when for every patch in the image its queries are aligned with the keys of a small number of other patches connected by the attention map," It is not clear what does this mean in a mathematical sense. There is also no detailed algorithmic description about how the model is trained. For example, for the image reconstruction task, how does minimizing the energy functions correspond to reconstruction? It would be helpful if the authors include a formal algorithmic description of the proposed method.

I am also concerned about some of the experiment settings. For example, for image reconstruction task, Figure 3 only shows some of the examples. However, this does not represent its general performance. One way to make this experiment more complete may be to include the MSE of the test datasets, the variance of the MSE, and perhaps reconstructed images corresponding to the best MSE and the worst MSE.


**Summary Of The Paper:**

This paper proposes the energy transformer, a transformer architecture that uses a recurrent energy transformer block. The energy transformer block updates its input in accordance to minimizing two energy functions. Experiments on image reconstruction and graph anomaly detection demonstrate the effectiveness of the proposed method.

**Summary Of The Review:**

Although the paper is interesting to read, I have two major concerns as detailed in the weakness section.

---

> ### Author Response · Authors · 2022-11-19
> **Response to Reviewer iuLg**
>
>
> Thank you for your comments and feedback. We are happy to hear that the main idea of  reformulating the transformer architecture in a way to combine Hopfield networks and attention networks, which are recurrently optimized using two carefully designed energy functions, got across! We respond to each question below:
>
> > "The log-sum energy function (3) is minimal when for every patch in the image its queries are aligned with the keys of a small number of other patches connected by the attention map," It is not clear what does this mean in a mathematical sense.
> >
>
> Consider, for simplicity, the situation when there is only one head. Then equation (3) reduces to
>
> $$
> E{^\text{ATT}} = -\frac{1}{\beta}\sum\limits_C \textrm{log} \left(\sum\limits_{B \neq C} \textrm{exp}\left(\beta \sum\limits_{\alpha} K_{ \alpha B}  Q_{\alpha C}\right) \right)
> $$
>
> Since exponential function grows very fast, in the sum over $B$ only a few terms will dominate (these large terms correspond to keys that have large overlaps with the queries for token $C$). Since $\log(\cdot)$ is an increasing function, large overlaps will result in large values of the logarithm. Because of the overall minus sign, this would lead to low energy states for such terms with large overlaps. For example, consider a simple situation when there are only two patches: $B,C=1,2$. Then, the above expression reduces to
>
> $$
> E{^\text{ATT}} = -\frac{1}{\beta}\log\Big( \textrm{exp}\big(\beta \sum\limits_{\alpha} K_{ \alpha 2}  Q_{\alpha 1}\big)\Big)  -\frac{1}{\beta}\log\Big( \textrm{exp}\big(\beta \sum\limits_{\alpha} K_{ \alpha 1}  Q_{\alpha 2}\big)\Big) = - \sum\limits_{\alpha} K_{ \alpha 2}  Q_{\alpha 1} - \sum\limits_{\alpha} K_{ \alpha 1}  Q_{\alpha 2}
> $$
>
> This energy is minimal (large in absolute value and negative) when both the overlaps of query of token 1 (or 2) with the key of token 2 (or 1) is large and positive.
>
> > No detailed algorithmic description about how the model is trained. For the image reconstruction task, how does minimizing the energy functions correspond to reconstruction? It would be helpful if the authors include a formal algorithmic description of the proposed method.
>
> The main idea of our approach is that the attention energy assigns low energy states to the plausible configurations of how different image patches can be put together in one image. For instance, in the example above patch 1 needs to agree with patch 2 on what is plotted in both of them in order to achieve low energy. In other words, commonly occurring "rules" of assembling the realistic images will have low energy. Examples of such rules: straight lines tend to continue in space, masked token in the middle of the blue sky it likely to be blue as well, etc.
>
> Thanks for the suggestion, we have added the formal algorithmic description to Appendix G. It provides step by step instructions on how the models have been trained and used at inference time.
>
> > Figure 3 only shows some of the examples. However, this does not represent its general performance. One way to make this experiment more complete may be to include the MSE of the test datasets, the variance of the MSE, and perhaps reconstructed images corresponding to the best MSE and the worst MSE.
>
> Thank you for the suggestion. We have included this analysis in Appendix D, see also Figure 9 and Figure 10.

---

> > ### Comment · Reviewer_iuLg · 2022-12-06
> > **I Appreciate the Response and Efforts**
> >
> > I would like to thank the authors for answering my questions and concerns. With the updated details, the paper is clearer to me, and I would like to raise my score towards acceptance.

---

### Author Response · Authors · 2022-11-19
**General Response**

Dear Reviewers, thank you very much for your valuable comments and feedback. Here is the summary of the main changes that we have made:

- We have added ablation studies that tackle the impact of subnetworks of ET on the final performance. Also, the impact of including the self-attention terms in the attention energy. Please see Appendix D.
- We have added results on the classification of graphs, Appendix F.
- We have quantified the variability and best/worst cases for the image reconstruction task, Appendix D2.
- The results for anomaly detection in heterogeneous setting are added to Appendix E.
- A formal algorithmic description of the training and inference pipelines is added to Appendix G.
- To help reproducibility, we add the [anonymized link](https://anonymous.4open.science/r/energy-transformer-19FF) to our model implemented in JAX.

We also address the questions raised in the individual responses to each reviewer, please see below.

We want to emphasize that the main contribution of our work is the novel **theoretical idea that the operation of the transformer can be conceptualized as a computation performed by a single large associative memory model**. As we explain in the response to Reviewer VTZR, this has never been done before. We believe that this main contribution is of great interest for the ML community. In addition to this, we show strong empirical results of our model on image completion, graph anomaly detection, and also on graph classification. These results go a long way to address reviewer suggestions for more results on graph and image tasks. Nevertheless, we leave NLP tasks for future work, given that the main contribution is to present a promising new model architecture and some of its applications. We hope that the additional results and clarifications address most of the questions raised by the reviewers.

---

### Decision · Program_Chairs · 2023-01-20

**Decision:**

Reject

**Justification For Why Not Higher Score:**

The paper only tests the ET method on a graph-based problem and lacks empirical evidence on computer vision and NLP, which are more commonly used and important domains for transformers.

**Justification For Why Not Lower Score:**

N/A

**Metareview: Summary, Strengths And Weaknesses:**

Inspired by the Hopfield Networks and Associative Memory model, the paper proposes a new transformer architecture called energy transformer (ET) .The sequence of transformer layers in ET is purposely designed to minimize a handcrafted energy function, which captures the relationships between the tokens. Empirical results are shown to demonstrate the effectiveness of the proposed framework. After the rebuttal, most of the reviewers found the presented energy transformer new, interesting and promising, and leant to accept the paper. However, Reviewer v3Fm and VTZR think that the current experiments are insufficient to demonstrate the effectiveness of the model because the paper is not a theoretical work and extensive experimental results should be necessary to justify the idea.   Reviewer v3Fm points out that the paper only tested the ET method on a graph-based problem and lacks empirical evidence on computer vision and NLP, which are more commonly used and important domains for transformers. After an internal discussion with reviewers, the AC thinks that the research will be of interest to the community, but the current paper is not yet ready for publication because of the insufficient and incomplete empirical evidence for verifying the method. The AC thinks that Reviewer v3Fm's concern is reasonable, and the proposed architecture needs to be tested thoroughly to demonstrate its performance before drawing safe and reliable conclusion. The AC recommends rejecting the paper at the current stage, and urges the authors to improve their paper by taking into account all the suggestions from the reviewers, and re-submit the paper in the next venue.


**Summary Of Ac-Reviewer Meeting:**

Reviewer v3Fm points out that the paper only tested the ET method on a graph-based problem and lacks empirical evidence on computer vision and NLP, which are more commonly used and important domains for transformers.

Reviewer VTZR lean to accept it because you think the paper have proved the concept.

Even though there is a divergence between the recommendations, they are based on the same facts in the paper. That is, (1) The idea is new and interesting, (2) the current paper is a non-theoretical paper, and (3) the current experiment results look promising, but neither thorough nor extensive.

The AC agrees with Reviewer v3Fm's opinion because the current paper is not a theoretical paper so that standard and extensive experiments are necessary to convince readers and demonstrate the design.

The AC recommends rejecting the paper at the current stage.